



# Deep-sea sponge grounds as nutrient sinks: High denitrification rates in boreo-arctic sponges

Christine Rooks[1], James Kar-Hei Fang[2], Pål Tore Mørkved[3], Rui Zhao[1], Hans Tore Rapp[1, 4, 5],

Joana R. Xavier[1, 6] and Friederike Hoffmann[1].

[1]Department of Biological Sciences, University of Bergen, Postboks 7803, 5020, Bergen, Norway.

[2]Department of Applied Biology and Chemical Technology, The Hong Kong Polytechnic University, Hung Hom, Hong Kong.

[3]Department of Earth Sciences, University of Bergen, Postboks 7803, 5020, Bergen, Norway.

[4]K.G. Jebsen Centre for Deep Sea Research, University of Bergen, Postboks 7803, 5020, Bergen, Norway.

[5]NORCE, Norwegian Research Centre, NORCE Environment, Nygårdsgaten 112, 5008 Bergen, Norway

[6]CIIMAR – Interdisciplinary Centre of Marine and Environmental Research of the University of Porto, 4450-208 Matosinhos, Portugal.

**Key words**: Denitrification, nitrification, boreal, Arctic, deep-sea sponges, sponge grounds.



**Abstract**

Sponges are commonly known as general nutrient providers for the marine ecosystem, recycling organic matter into various forms of bio-available nutrients such as ammonium and nitrate. In this study we challenge this view. We show that nutrient removal through microbial denitrification is a

common feature in six cold-water sponge species from boreal and Arctic sponge grounds. Denitrification rates were quantified by incubating sponge tissue sections with $^{15}NO_3^-$ - amended oxygen saturated seawater, mimicking conditions in pumping sponges, and de-oxygenated seawater, mimicking non-pumping sponges. Rates of anaerobic ammonium oxidation (anammox) using incubations with $^{15}NH_4^+$ could not be detected. Denitrification rates of the different sponge

species ranged from 0 to 114 nmol N $cm^{-3}$ sponge $day^{-1}$ under oxic conditions, and from 47 to 342 nmol N $cm^{-3}$ sponge $day^{-1}$ under anoxic conditions.

An exponential relationship between the highest potential rates of denitrification (in the absence of oxygen) and the species-specific abundances of *nir*S and *nir*K genes encoding nitrite reductase, a key enzyme for denitrification, suggests that the denitrifying community in these sponge species is

both prepared and optimized for denitrification. The lack of a lag phase in the linear accumulation of the $^{15}N$ labelled $N_2$ gas in any of our tissue incubations is another indicator for an active community of denitrifiers in the investigated sponge species.

High rates for coupled nitrification-denitrification (up to 89% of nitrate reduction in the presence of oxygen) shows that under these conditions, the $NO_3^-$ reduced in denitrification was primarily

derived from nitrification within the sponge, directly coupling organic matter degradation and nitrification to denitrification in sponge tissues. Under anoxic condition when nitrification was not possible, nitrate to fuel the much higher denitrification rates had to be retrieved directly from the seawater. The lack of *nif*H genes encoding nitrogenase, the key enzyme for nitrogen fixation, shows



that the nitrogen cycle is not closed in the sponge grounds. The denitrified nitrogen, no matter of its origin, is then no longer available as a nutrient for the marine ecosystem.

Considering average sponge biomasses on typical boreal and Arctic sponge grounds, our sponge denitrification rates reveal areal denitrification rates of 0.8 mmol N $m^{-2}$ $day^{-1}$ assuming non-pumping sponges and still 0.3 mmol N $m^{-2}$ $day^{-1}$ assuming pumping sponges. This is well within the range of denitrification rates of continental shelf sediments. For the most densely populated boreal sponge grounds we calculated denitrification rates of up to 2 mmol N $m^{-2}$ $day^{-1}$, which is comparable to rates in coastal sediments. Increased future impact of sponge grounds by anthropogenic stressors reducing sponge pumping activity and further stimulating sponge anaerobic processes may thus lead to that deep-sea sponge grounds change their role in the marine ecosystem from nutrient sources to nutrient sinks.





## 1    Introduction

Sponges are sessile filter feeders with an immense capacity to process large volumes of seawater (Kahn et al., 2015;Reiswig, 1974). As such, they play a critical role in benthic-pelagic coupling, recycling particulate or dissolved organic matter from the water column into various forms of bio-available nutrients (Brusca and Brusca;Reiswig, 1974;Yahel et al., 2003;Hoffmann et al., 2009;Schläppy et al., 2010a;Maldonado et al., 2012;de Goeij et al., 2013;Rix et al., 2016). Actively pumping sponges have been associated with the release of dissolved inorganic nitrogen (DIN), enriching ex-current waters with excess $NH_4^+$ and or $NO_x^-$ ($NO_3^-$ and or $NO_2^-$) (Southwell et al., 2008;Fiore et al., 2013;Keesing et al., 2013;Leys et al., 2018;Hoer et al., 2018). Whilst $NH_4^+$ is excreted by sponge cells as a metabolic waste product (Yahel et al., 2003), $NO_x^-$ is derived from the microbial oxidation of $NH_4^+$, through $NO_2^-$, to $NO_3^-$ in aerobic nitrification (Painter, 1970;Corredor et al., 1988;Diaz and Ward, 1997;Jiménez and Ribes, 2007;Schläppy et al., 2010a;Southwell et al., 2008;Radax et al., 2012;Fiore et al., 2010).

Nitrogen fixation has also been reported in shallow water sponges (Wilkinson and Fay, 1979;Wilkinson et al., 1999;Mohamed et al., 2008;Ribes et al., 2015), reducing biologically inaccessible $N_2$ gas to $NH_4^+$. Although this pathway represents yet another source of bio-available N, the presence of *nifH* (encoding the essential nitrogenase responsible for $N_2$ fixation) does not necessarily confer to nitrogen fixing activity, even under seasonal N-limitation (Ribes et al., 2015;Bentzon-Tilia et al., 2014). DIN release has been affiliated with a number of deep-sea and shallow water sponges and varies according to species (Schläppy et al., 2010a;Radax et al., 2012;Keesing et al., 2013), as well as on temporal (Bayer et al., 2008;Radax et al., 2012) and spatial scales (Fiore et al., 2013;Archer et al., 2017). Such variations have been linked to abiotic conditions and the availability of N-rich particulate organic matter in the water column (Bayer et al., 2008;Archer et al., 2017;Fiore et al., 2013), although where archaea dominate the nitrifying

community, $NO_3^-$ release could potentially be sustained by nitrifying archaea scavenging for $NH_4^+$ under N-limiting conditions (Radax et al., 2012;Tian et al., 2016)

In any case, $NO_3^-$ release is dependent on active filtration, delivering an excess of $O_2$ to sponge tissues, and in turn, sustaining aerobic nitrification within the sponge (Reiswig, 1974;Hoffmann et al., 2008;Southwell et al., 2008;Pfannkuchen et al., 2009;Fiore et al., 2013;Keesing et al., 2013;Leys et al., 2018). Fluctuations in pumping activity, however, disrupt the delivery of $O_2$ to sponge tissues, resulting in either heterogeneous oxygenation within the sponge matrix or complete anoxia (Hoffmann et al., 2005;Hoffmann et al., 2008;Schläppy et al., 2010b;Schläppy et al., 2007). Under such conditions, a paucity of oxygen would inevitably promote anaerobic microbial processes.

Anaerobic N-transformations have been quantified using $^{15}N$ tracer experiments in deep-sea (Hoffmann et al., 2009) and shallow water sponges (Schläppy et al., 2010a;Fiore et al., 2013). In the deep-sea sponge, *Geodia barretti,* the removal of fixed nitrogen via heterotrophic denitrification (the sequential and anaerobic reduction of $NO_3^-$, via $NO_2^-$, to $N_2$) was shown to exceed sedimentary denitrification rates at equivalent depths by a factor 2 to 10 (Hoffmann et al., 2009). Given that marine sediments are considered the major sites of marine N-transformations (Middelburg et al., 1996;Seitzinger, 1988), sponges may thus represent a significant, yet largely overlooked sink for bioavailable nitrogen (Hoffmann et al., 2009).

This opens for an alternative role of sponges as nutrient scavengers, and an alternative explanation for the observed variations in nutrient release by sponges: a combination of variations in remineralisation processes associated with food availability **and** direct consumption of endogenous



and ambient nutrients by microbial processes in sponges. The balance of these processes and their controlling factors, however, have not as yet been quantified.

The understanding of such processes and their dynamics is particularly relevant for areas where sponges occur in high densities forming highly structured habitats as is the case of the sponge grounds found widely distributed across the deeper areas of the oceans. In such areas, sponges can represent up to 95% of the total invertebrate biomass (Murillo et al., 2012) and attain densities of up to 20 individuals m$^{-2}$ (Hughes and Gage, 2004). *Geodia barretti* is a key species on boreo-arctic sponge grounds, which cover wide expanses of the seafloor both in the Eastern and Western North Atlantic, as well as the sub-Arctic (Klitgaard and Tendal, 2004;Murillo et al., 2012). However, to make reliable estimates on the potential nitrogen sink function of these deep-sea sponge grounds, denitrification rates from more sponge ground species are needed.

In this study we quantify the potential nutrient sink function of six sponge species which characterize the two main types of boreo-arctic Tetractinellid sponge grounds. We aim to show that nutrient removal through microbial denitrification is a common feature in cold-water sponges, and that rates are dependent on oxygen availability in the sponge tissue. Based on these results we aim to estimate the potential nutrient sink function of boreo-arctic sponge grounds for the marine ecosystem.



## 2 Materials and Methods

### 2.1 Site description

Arctic sponge species were collected at the Schulz Bank (73° 50' N, 7° 34' E). This is a large seamount located at the transition between the Mohn and the Knipovich ridges, two of the main

sections of the Arctic Mid-Ocean Ridge (AMOR). The seamount rises from more than 2.500 m depth and its summit and shallower areas (550-700 m depth) host a dense and diverse sponge ground composed of a multispecific assemblage of species dominated by tetractinellid desmosponges (*Geodia parva*, *G. hentscheli*, and *Stelletta rhaphidiophora*) and hexactinellid sponges (*Schaudinnia rosea*, *Trichasterina borealis*, *Scyphidium septentrionale*, and *Asconema*

*foliata*). Exact hydrodynamic settings at the summit is not known, but conditions measured using a benthic lander at 670 m (i.e. 70-80 m below it) revealed a water temperature just below 0 °C, salinity of 34.9, and dissolved oxygen between 12.4-12.6 mg L$^{-1}$. Near-bed suspended particulate matter concentrations was determined to be 3.2 mg L$^{-1}$, considerably larger than those observed both in surface and deeper waters (where values range from less than 1 and 2 mg L$^{-1}$) (Roberts

and J., 2018;Roberts et al., 2018).

Boreal sponge species were collected on the hard bottom slope of the fjord Korsfjord (60°09′12″N, 05°08′52″E) near the city of Bergen on the west coast of Norway. Hard bottom slopes of these fjords, which can be several hundred meters in deep, host dense assemblies of typical boreal sponges, dominated by tetractinellid demosponges such as different species of the

Geodiidae. Site characteristics are described elsewhere (Hoffmann et al., 2003).
Average sponge biomass (kg/m$^2$) in both Arctic and boreal grounds was estimated from trawl catches and underwater imagery collected in the course of various sampling campaigns.





### 2.2 Sample collection and preparation

Intact individuals from each of the key Arctic species, *Geodia hentscheli* (n=3), *Geodia parva* (n=3) and *Stelletta rhaphidiophora* (n=3) were retrieved from a depth of 700m at the top of Schulz Bank. Sponges were collected with a remotely operated vehicle (ROV) on board the R/V GO Sars

in June 2016

Intact individuals from each of the key boreal species, *Geodia barretti* (n=3), *Geodia atlantica* (n=3) and *Stryphnus fortis* (n=3) were collected from a depth of 200m at the slope of Korsfjorden, Norway. Sponge individuals were retrieved using a triangular dredge deployed from the R/V Hans Brattstrøm in November 2016. As sponges were collected at the rocky slope of the fjord, it was not

possible to collect sediment from that sampling site.

Upon retrieval, samples were immediately transferred into containers holding low-nutrient seawater, directly recovered from the sampling site. Following species identification, intact individuals were either transported to the aquaria at the University of Bergen (ca. 1h; boreal species), or immediately to the lab on board the R/V G.O. SARS (Arctic species). Sponge tissue,

from three intact individuals, was then dissected for use in either $^{15}$N-labelled tissue incubations or preserved for subsequent DNA extraction for each species.

Whilst completely immersed in site water, sponge individuals were cut into 3 sections of approximately equal size. Using an autoclaved stainless steel core (internal diameter = 0.74cm; length = 7cm), the choanosomal portion of the sponge was sliced from each section to produce

cylindrically-shaped tissue samples. Avoiding exposure to air, tissue samples were then transferred to 1L containers holding site water. Using a sterile scalpel, the tissue cylinders were further sectioned (under water) into pieces of equal size (volume = 0.45cm$^3$). The samples were then either

distributed into 12mL gas-tight vials (Exetainer, Labco, High Wycombe, UK) for incubation with

$^{15}N$ isotopes, or into 1.5mL microcentrifuge tubes, snap frozen and stored at -80°C for subsequent

DNA extraction.

Sediment was collected from the Arctic sponge grounds using a box-corer. The upper few

centimeters were sampled, homogenised and packed into 10 mL sterile cut-off syringes. 1mL of

sediment was then either distributed into 3mL gas tight vials (Exetainer, Labco, High Wycombe,

UK) for $^{15}N$ isotope incubations or into 1.5ml microcentrifuge tubes (Eppendorf), snap-frozen and

stored at -80°C for subsequent DNA extraction.

### 2.3 Quantifying rates of N-removal processes in sponge tissues and deep-sea sediments

2.3.1 Sponge tissue incubations

For simulating conditions in pumping and non-pumping sponges, sponge tissue sections were

incubated with oxygen-saturated (standard temperature and pressure) and degassed site water

(oxygen free seawater, degassed with ultra high purity He). Site water was retrieved using 10L

Niskin flasks mounted on a CTD rosette water sampler aboard the R/V GO Sars. This water was

collected at a depth of approximately 650m, just above the summit of the seamount. It was then

filtered to remove water column bacteria and or phytoplankton (0.2µm polycarbonate filters,

Whatman Nucleopore) and added to all incubations with Arctic specimens. Boreal specimens were

incubated with sand filtered seawater, pumped into the aquaria at the University of Bergen from a

local fjord. This water was sourced from a depth of 130m.

To ensure that all labelled $N_2$ gas was retained, it was necessary to maintain a gas-tight atmosphere

in each of the incubations. Consequently, no oxygen could be added during the experiment.

Estimating from typical respiration rates of 0.32 µmol $O_2$ mL sponge$^{-1}$ h$^{-1}$ in *G. barretti* (Leys et

al., 2018), this would suggest the complete removal of oxygen (by sponge cells and associated microbes) following 26 hours of incubation (12 ml exetainer, sponge pieces 0.45 cm$^3$, oxygen concentration at experiment start 313 µmol/L). This means that oxygen concentrations in the aerobic incubation continuously decreased from oxygen saturation to zero throughout the course

of the experiment, thus mimicking conditions where a sponge has recently ceased pumping, or where pumping occurs at a low rate (Fang et al., 2018;Hoffmann et al., 2008;Schläppy et al., 2010b). Nevertheless, we can assume that oxygen was available during the first 26 hours of incubation in the oxic experiment, in contrast to the anoxic experiment where oxygen was absent from the beginning of the incubation, thus mimicking non-pumping conditions (Hoffmann et al.,

2008;Schläppy et al., 2010b).

For the oxic incubations, 12 mL of air-saturated (standard temperature and pressure) seawater was transferred into 12 mL gas tight vials. Using autoclaved forceps, one piece of freshly dissected tissue was then placed into each gas tight vial, until a sufficient number of samples were prepared for the incubations. The caps were then replaced and the vial was carefully sealed to exclude any

air bubbles.

For the anoxic incubations, 2L of surface site water was de-gassed with ultra high purity He for 2h. To verify the absence of oxygen in the de-gassed water, an anaerob strip test (colour change from pink to white under anaerobic conditions; Sigma Aldrich) was performed prior to transfer into 12mL exetainers. The caps were then replaced and the gas tight vials were carefully sealed to

exclude any air bubbles.

Incubations were prepared in four sets of 1 un-amended reference (no isotope added) and 5 amended ($^{15}$N labelled) samples per in-tact sponge (x3 in-tact sponge individuals/ species). Each

set was then either injected (gas tight luer lock syringes, VICI, USA) with air saturated (at standard

temperature and pressure, for oxic incubations) or oxygen free (de-gassed; for anoxic incubations)

concentrated stock solutions of i) $Na^{15}NO_3^-$ (99.2 $^{15}N$ atm. %); $^{14}NH_4^+$ $Cl^-$ or ii) $^{15}NH_4^+$ $Cl^-$ (≥98.

$^{15}N$ atm %); $Na^{14}NO_3^-$ and shaken vigorously. The final concentrations of i) $^{15}NO_3^-$; $^{14}NH_4^+$

(screening for denitrification and/or anammox) or ii) $^{15}NH_4^+$; $^{14}NO_3^-$ were 100 µM $NO_3^-$ and 10 µM

$NH_4^+$ respectively. These values were essentially 90% above ambient $NO_3^-$ (10 µM $NO_3^-$) and $NH_4^+$

concentrations (<1 µM $NH_4^+$) present in the seawater. Prior to the incubations, however,

background nutrient concentrations were unknown. In this regard, to ensure that the availability of

$^{15}N$ was sufficient for the measurement of denitrification and or anammox (e.g. at least 50% above

the ambient pool of $^{14}N$), we selected high concentrations of stock solutions (Holtappels et al.,

2011). To enable continuous homogenisation of the isotopic label with sponge tissue, exetainers

were placed on rollers (Spiromix, Denley) and incubated at in situ temperature (6°C) in the dark.

At zero hours, and at subsequent 3-6 hour intervals, a selection of samples were injected with 2mL

of ultra high purity helium to create an oxygen free headspace using a gas-tight syringe. The vials

were then injected with 200µL of formaldehyde, and shaken vigorously to inhibit further microbial

activity. This was repeated over a period of 48 hours.

### 2.3.2 Sediment slurry incubations

One mL of the homogenised sediment was distributed into 3mL gas-tight vials (Exetainer, Labco,

High Wycombe, UK) with 1mL of de-gassed site water (as above). The cap was replaced, the

headspace (1mL) flushed with ultra-high purity helium and each vial was shaken vigorously to

produce an anaerobic sediment slurry. Anaerobic slurries were prepared as 2 sets of un-amended

references (no isotopic mixture added) and 5 amended samples in incubations screening for either

anammox and or denitrification. Amended samples were injected with oxygen free isotopic

mixtures (as above) and placed on rotating rollers (Spiromix, Denley) in a constant temperature

room (6°C) in the dark. At zero hours, and every subsequent 3-6 hours, a selection of samples was

injected with 200µL of formaldehyde, and shaken vigorously to inhibit further microbial activity.

This was repeated over a period of 48 hours. Concentrations of $^{28}N_2$, $^{29}N_2$ and $^{30}N_2$ were measured

as above and calculations for denitrification and or anammox were performed as per (Thamdrup

and Dalsgaard, 2002)  and (Risgaard-Petersen et al., 2003).

*2.3.3 Calculation of denitrification and anammox rates*

Concentrations of $^{28}N_2$, $^{29}N_2$ and $^{30}N_2$ were measured by directly sub-sampling 70µL from the gas

headspace on a GC (Trace GC, Thermo Fisher Scientific, Bremen) connected to a continuous flow

isotope ratio mass spectrometer (Delta V plus, Thermo Fisher Scientific, Bremen) calibrated with

in house reference gas and air. Calculations for rates of both anammox and denitrification were

based on established methods for measuring these processes in sediments (Thamdrup and

Dalsgaard, 2002;Risgaard-Petersen et al., 2003). Rates were calculated from the linear increase in

the $N_2$-accumulation over time as measured from the isotope ratio mass spectrometer.

The accumulation of excess $^{29}N_2$ and $^{30}N_2$, from incubations with $^{15}NO_3^-/^{14}NH_4^{+-}$ , was linear over

a 24h period ($p<0.05$) and precluded an initial lag phase (Figures 1a and 1b). This was the case

for all species. In the oxic incubations, after 24 hours a sharp non-linear increase in labelled $N_2$

was detected. This is in good agreement with our calculations for oxygen depletion (26 hours, see

above). Since we observed no signs of tissue degradation during the 48 hours of incubation (e.g.



tissue pieces turn black, see for example (Hoffmann et al., 2003;Osinga et al., 2001;Osinga et al., 1999), this non-linear increase was taken to indicate a switch of metabolic processes within the sponge towards predominantly anaerobic pathways, and thus, a different denitrification rate. For the anoxic incubations, $N_2$-production was also linear during the first 24 hours of incubations,

although the data were more scattered when compared with oxic incubations. The scatter increased after 24 hours, though most incubations still followed a similar linear trend. Also here, no signs for tissue degradation were observed.

For best comparability of denitrification rates from oxic and anoxic incubations, only the first 24 hours, where $N_2$ production was linear in all experiments, and where oxygen was assumed to be

present in the exetainers of the oxic incubation, were used to calculate denitrification rates.

No $^{29}N_2$ production was detected following labelling with $^{15}NH_4^+$ and $^{14}NO_3^-$, suggesting an absence of anammox activity. Therefore, no anammox rates could be calculated. The $N_2$ produced during the $^{15}NO_3^-/^{14}NH_4^+$ experiments is assumed to originate entirely from denitrification.

*2.3.4 Calculation of coupled nitrification-denitrification and the denitrification of $NO_3^-$ derived from ambient seawater*

To determine the predominant source of $NO_3^-$ fueling denitrification, rates of coupled nitrification-denitrification and the denitrification of $NO_3^-$ supplied by the ambient seawater, were calculated according to the methods of Nielsen (1992) (Nielsen, 1992). In brief, the production of $NO_3^-$ can

occur endogenously via the aerobic oxidation of $NH_4$ to $NO_3^-$ within the sponge tissues. In turn, this represents a source of $NO_3^-$ for denitrification which 'couples' nitrification to denitrification. Alternatively, denitrification can simply be fueled by $NO_3^-$ diffusing from the ambient seawater. By taking into consideration the frequency of $^{14}$ and $^{15}NO_3^-$ availability, in addition to random

isotope pairing, it is possible to calculate the source of denitrified $NO_3^-$ from the abundance of [28],

[29] and $^{30}N_2$ in all oxic incubations.

## 2.4 Screening and quantifying the abundance of *nir*S, *nir*K and *nif*H genes

Total DNA was extracted from dissected sponge pieces (0.45cm$^3$of sponge tissue) using a

FastDNA Spin Kit for Soil (mpbio, Santa Ana, CA, USA) following the manufacturer's

instructions. In total, DNA was extracted from 3 tissue samples retrieved from each of the intact

sponges (3 intact individuals sampled/ key species) as well sediment samples (1mL, ~2g sediment

slurry) and sample blanks (RNAse free water). DNA extracts were eluted into 100 μL of PCR grade

double distilled $H_2O$ and stored at -20°C until further analysis.

The functional genes diagnostic of nitrogen fixation (*nif*H encoding nitrogenase) and denitrification

(*nir*S/K encoding nitrite reductase) in sponges were screened using conventional PCR of 40 cycles.

*nif*H gene was amplified using the primer pair nifHfw/nifHrv  (Mehta et al., 2003) with the

following thermal conditions: 94°C for 15 min, and 40 cycles of 94°C for 30 s, 55°C for 30 s, 72°C

for 60 s. nirS/K genes were amplified using the primers and thermal conditions as described below.

Each reaction mixture (25μl total volume) contained the following: 1× HotStar Taq® Master Mix

(Qiagen, Hilden, Germany), 1.2 μM of each primer and 1 μl template DNA. PCR products were

evaluated by visual inspection on 1% agarose gels.

The abundance of *nir*S or *nir*K genes of denitrifying bacteria were quantified using quantitative

PCR (qPCR) on a StepOne Real-Time PCR system (Applied Biosystems). *nir*S genes were

amplified using the primer pair nirS_cd3aF/nirS_R3cd (Throback et al., 2004), with thermal

conditions as follows: 95°C for 15 min, 45 cycles of denaturing at 95°C for 15 s, annealing at 51°C



for 30 s, and elongation at 72°C for 45 s. The *nirK* gene was amplified using the primer pair nirK_F1aCu/ nirK_R3Cu, with the following thermal conditions: 95°C for 15 min, 45 cycles of denaturing at 95°C for 30 s, annealing at 51°C for 45 s, and elongation at 72°C for 45 s. All qPCR reactions were run in triplicate and each reaction mixture contained 1× QuantiTech SybrGreen

PCR master mixture (QIAgen, Germany), 0.5 µM forward and reverse primer and 1 µl of DNA template in a final volume of 20 µL. Standard of qPCR of each gene was linear DNA containing respective genes from an uncultured denitrifying bacterium in an Arctic permafrost soil. For each gene, the DNA concentration of the standard was measured using BIO-analyzer (DNA 1000 chips, Agilent Technologies) and a DNA abundance gradient of $10$-$10^5$ copies $\mu L^{-1}$ were prepared by 10x

serial dilution.

## 2.5 Statistical analyses

Statistical analyses were performed to test for significant differences in (i) species-specific rates of denitrification or (ii) variations in the rates of denitrification according to oxygen availability. The

data set failed to meet the assumptions of normality or equal variance. As a result, the data set wastransformed by rank prior to two-way ANOVA. All pairwise multiple comparisons were performed using the Holm-Sidak method at species level. In all cases, the level of significance was set to at least $p < 0.05$. Statistical analyses were performed using the software SigmaPlot 13.0 (Systat Software, CA, USA).





## 3 Results

### 3.1 Denitrification activity in sponge tissues

$N_2$ production via anammox requires 1 N from $NO_3^-$ (which is $^{15}N$ labelled) and 1 N from $NH_4^+$ (which is not labelled). The lack of $^{29}N_2$ production following labelling with $^{15}NH_4^+$ as observed in

our study suggests an absence of anammox. Therefore, no anammox rates could be calculated and the labelled $N_2$ produced during the $^{15}NO_3^-$ incubations is assumed to originate entirely from denitrification. Denitrification rates as calculated from this linear $N_2$-release were quantified in all 6 sponge species and are shown in figure 2. Mean rates of denitrification varied significantly between species (two-way ANOVA, $F_{1,5}=39.339$, $p<0.01$) and in the presence or absence of

dissolved oxygen (two-way ANOVA, $F_{1,5}=50.260$, $p<0.01$). A significant interaction between species and the availability of dissolved oxygen was also identified by two-way ANOVA ($F_{1,5}=2.847$, $p=0.037$). Mean rates of denitrification were always greater in incubations with de-gassed seawater relative to incubations with fully air saturated seawater (Fig. 2). Under oxic conditions, mean rates varied from 0 nmol N cm$^{-3}$ sponge day$^{-1}$ in *Stryphnus fortis* to a maximum

of 114 nmol N cm$^{-3}$ sponge day$^{-1}$ in *Stelletta rhaphidiophora*. However, under anoxic conditions, rates of denitrification ranged from 47 nmol N cm$^{-3}$ sponge day$^{-1}$ in *Stryphnus fortis* to 342 nmol N cm$^{-3}$ sponge day$^{-1}$ in *Geodia parva* (Fig.2.). Differences in the rates of denitrification under either aerobic or anaerobic conditions were significant in *Stryphnus fortis* ($t=2.075$, $p<0.05$), *Geodia barretti* ($t=3.277$, $p<0.05$), *Geodia hentscheli* ($t=3.495$, $p<0.05$) and *Geodia parva* ($t=5.789$,

$p<0.05$). Notably, the Arctic sponge ground species *G. hentscheli* and *G. parva* showed the highest anaerobic denitrification rates.

No labelled $N_2$ production was detected in the surface sediment slurries screening for denitrification or anammox.





**3.2 Coupled nitrification-denitrification and the absence of nitrogen fixation**

In incubations with air saturated seawater, denitrifying activity was detected in all sponges with the exception of *Stryphnus fortis* (Fig. 2). For these species 67 – 89% of the nitrate reduced in denitrification was coupled to nitrification within the sponge tissue. This indicates a minimum

nitrification rate of 29 – 102 nmol N cm$^{-3}$ sponge day$^{-1}$ and the predominance of this process as a source of nitrate for denitrification under oxic conditions (Table 1).

Functional genes for nitrogen fixation were not detected in any of the six sponge species, pointing towards the absence of nitrogen fixing microorganisms in these species.

**3.3 Correlation between denitrification rates and the abundance of nitrite reductase**

Copies of the nitrite reductase genes, *nir*S and *nir*K, were detected in all six sponges, though in different quantities (Table 2). The total nitrite reductase copy number (the sum of mean *nir*S and *nir*K gene copies per cm$^{-3}$ sponge tissue) ranged from 2.19E+03 copies cm$^{-3}$ sponge in *Stryphnus fortis* to 1.03E+09 copies cm$^{-3}$ sponge in *Geodia parva* (Table 2). Although no denitrification

activity was measured in the sediment slurry incubations, nitrite reductase was present at an abundance of 2.77E+04 copies cm$^{-3}$ sediment.

A positive exponential relationship between denitrification rates under anoxic conditions, and total *nir* copy number, were associated with 4 of the sponge species (*Stelletta rhaphidophora, Geodia barretti, G. hentscheli and G. parva*) with the highest anaerobic denitrification rates (>140 nmol N

cm$^{-3}$ sponge day$^{-1}$, Fig. 3.). No correlation, however, was detected for species with low denitrification rates under anoxic conditions (<60 nmol N cm$^{-3}$ sponge day$^{-1}$), nor for denitrification rates under oxic conditions, where the rates were quite similar for the 4 most active species despite variations in *nir* gene copy number (Fig. 3.).

25                                                   17



## 4 Discussion

### 4.1 Denitrification as a common feature of cold-water sponges

The purpose of this study was to quantify the potential nutrient sink function of six sponge key

species from boreal and arctic sponge grounds. **We aimed to show that (1) nutrient removal through microbial denitrification is a common feature in cold-water sponge species, and that (2) rates are dependent on oxygen availability in the sponge tissue.**

All six species investigated in this study showed denitrification rates under anoxic conditions, five of them even under oxic conditions. Rates were always higher in the absence compared to in the

presence of oxygen. All our denitrification rates are within the same range as rates previously reported for cold- and warm-water sponges: Hoffmann et al. (2009) reported 92 nmol N cm$^{-3}$ sponge day$^{-1}$ for explants of *G. barretti* incubated under oxic conditions, which is very close to our average rate of 86 nmol N cm$^{-3}$ sponge day$^{-1}$ for *G. barretti* sections incubated under oxic conditions. Rates reported by Schläppy et al. (2010a) for the two Mediterranean shallow water

sponges *Chondrosia reniformis* and *Dysidea avara,* also measured on tissue sections incubated under oxic conditions, were 240 and 357 nmol N cm$^{-3}$ sponge day$^{-1}$, respectively – well above our maximum rates measured under oxic conditions, but close to our rates measured under anoxic conditions (with 342 N cm$^{-3}$ sponge day$^{-1}$ for *G. parva* as our highest rate). Considering generally higher metabolisms in warm and shallow waters compared to cold deep waters, this is as expected.

In addition to these rather few quantifications of denitrification rates in sponges, the presence of denitrification activity has been shown by isotopic tracer experiments in a tropical sponge (Fiore et al., 2013), as well as by numerous reports on the presence of functional genes for denitrification in sponge microbes, or by proving the ability for denitrification in sponge-derived microbial isolates from a variety of marine habitats (Bayer et al., 2014;Cleary et al., 2015;Fiore et al.,



2010;Fiore et al., 2015;Han et al., 2013;Li et al., 2014;Liu et al., 2016;Liu et al., 2012;Webster and Taylor, 2012;Zhang et al., 2013;Zhuang et al., 2018).

We could not detect any anammox rates in any of the sponges investigated in this study. The only literature report for anammox rates quantified in a sponge was a very low rate of 3 nmol cm$^{-3}$

sponge day$^{-1}$ in explants of *G. barretti* (Hoffmann et al 2009). In the present study, we could not reproduce these rates in the tissue sections of *G. barretti* nor detect the functional genes associated with this process. There are no other quantifications of anammox rates in sponges, and only few studies on the presence of anammox bacteria and genes in some sponge species (Han et al., 2012;Mohamed et al., 2010;Webster and Taylor, 2012).

Our study further clearly shows that denitrification rates are generally higher under anoxic conditions. With denitrification being an anaerobic process, this is not surprising. More surprising is our detection of considerable denitrification rates (up to 114 nmol N cm$^{-3}$ sponge day$^{-1}$) when sponge tissue sections were incubated in oxygenated seawater. Furthermore, the high percentage of coupled nitrification/denitrification (up to 89.5% for *Stelletta rhaphidiophora*, the

sponge with the highest oxic denitrification rate!), proves that both aerobic and anaerobic processes happened in the sponge sections at the same time. Oxygen was assumed to be present in the experimental vial at least during the first 26 hours of the experiment, though continuously decreasing due to sponge respiration (see calculation in method section), but we do not have control over oxygen concentration in the sponge tissue pieces during the experiment. From

marine sediments, there are numerous studies reporting denitrification in bulk oxic conditions, either in anoxic microniches or under complete oxygenated conditions. e.g.  (Wilson, 1978) (Marchant et al., 2017;Robertson et al., 1995;Chen and Strous, 2013). For the present study, we do not know if denitrification actually happened in the presence of oxygen, in anoxic

microniches, which were present in the sponge tissue already at experiment start, or in tissue sections rapidly becoming anoxic while not continuously flushed with oxygen. Nevertheless since all these scenarios reflect the situation in a sponge which is pumping on a low rate or occasionally stops pumping (Hoffmann et al., 2008;Schläppy et al., 2010b;Schläppy et al., 2007),

which are typical features in sponges, we assume that our results are representative for sponges under normal conditions.

Our study further indicates significant differences in (anaerobic) denitrification rates between (some) sponge species, with two of the three Arctic species (*G. hentscheli* and *G. parva*) displaying the highest rates. Sampling coincided with a seasonal pulse of organic matter in the water column

above the Schulz Bank (this was measured on the 2016 cruise at the time of sampling). An increase in the availability of organic matter is known to stimulate denitrification (Devol, 2015) and could explain, to some extent, why rates are higher among two of the Arctic samples retrieved in June 2016, relative to boreal samples retrieved in November 2017. Another explanation is that there are simply species-specific differences in maximum denitrification rates, independent of sampling site

and time.

Our systematic screening of 6 cold-water sponge species, together with reports of denitrification activity from other sponge species all over the world and from different habitats (see above), **strengthens the view that denitrification is a common feature in many sponge species – both under oxygenated (pumping) and deoxygenated (non-pumping) tissue conditions, with rates**

**being highest when oxygen is absent.** Anammox in contrast seems to be a more rare and occasional feature in sponges, which may not have quantitative importance for sponge-mediated nitrogen cycling.



### 4.2 The fate of nitrogen in sponges

With the exception of *Stryphnus fortis*, denitrification was verified in the presence of dissolved oxygen across all species. High rates for coupled nitrification-denitrification shows that under these conditions, the $NO_3^-$ reduced in denitrification was primarily derived from nitrification, coupling aerobic ammonium oxidation to $NO_3^-$ reduction in sponge tissues. Mean values of coupled nitrification-denitrification indicate that nitrification fuels as much as 67-89% of $NO_3^-$ reduction (Table 1). This exceeds values previously reported for explants of *Geodia barretti*, which showed similar denitrification rates as *G. barretti* sections incubated in oxic seawater in this study (see above,), but where only 26% of N-loss was attributed to coupled nitrification-denitrification (Hoffmann et al., 2009). Those measurements were derived from sponge explants, which inherently lack an aquiferous system (Hoffmann et al., 2003, 2005, 2009). As such, the delivery of oxygen is entirely dependent on diffusion, which renders a large portion of the explant matrix permanently anoxic (<0.5-1mm below the surface; Hoffmann et al., 2005). Since nitrification requires oxygen, the high rates of coupled nitrification-denitrification for *G. barretti* and the majority of sponges investigated in the present study proves that (1) for incubations in oxygen-saturated seawater, more oxygen was present in the sponge tissue sections compared to sponge explants incubated under the same conditions and (2) nitrate derived from nitrification within the sponge tissue fueled most of the denitrification under oxic conditions.

The calculated rates of coupled nitrification-denitrification represent the minimum nitrification rates. Since we did not measure nitrate production in the current study, true nitrification rates may be even higher. Nevertheless, minimum nitrification rates calculated in this study of up to 102 nmol N cm$^{-3}$ sponge day$^{-1}$, are just below nitrification rates quantified in the cold-water species *Phakellia ventilabrum, Antho dichotoma, Geodia barretti* and *Stryphnus fortis* (120-1880 nmol N cm$^{-3}$



sponge day$^{-1}$; Radax et al., 2012; Fang et al., 2018; Hoffmann et al., 2009). Since the ammonium concentration in bottom seawater at our sampling sites is far too low (under detection limit of 1 µM NH$_4^+$) to fuel our calculated (and probably even underestimated) nitrification rates, ammonium needs to origin from organic nitrogen remineralized from organic matter by the sponge cells or by

5   heterotrophic sponge microbes. Under anoxic conditions, there is no nitrification, and nitrate to fuel the much higher denitrification rates has to be retrieved directly from the seawater. We did not detect any genes for nitrogen fixation; the N-cycle is not closed in the cold-water sponges. **The denitrified nitrogen, no matter of its origin, is no longer available as a nutrient and thus inevitably lost as a good and service for the marine ecosystem**.

### 4.3 The sponge microbial community is ready for denitrification

*Nir*S and *nir*K are functionally equivalent genes that code for the reduction of nitrite to nitric oxide, the first step towards the production of a gas in denitrification (Shapleigh, 2013). They occur as single copies within the denitrifying genome, which generally indicates that a single copy of nitrite

15   reductase (*nir*S or *nir*K) corresponds to one cell with the potential for denitrification. Copies of *nir*S and *nir*K were detected in all six sponge species, and also in the sediment (Table 2). Scattering denitrification rates against nitrite reductase copy numbers, revealed an exponential relationship between the highest denitrification rates (in the absence of oxygen) and the species-specific abundance of *nir*S and *nir*K (Fig. 3). This relationship suggests that the denitrifying community in

20   *G. parva, G. hentscheli, G. barretti* og *S. rhaphidophora* is both prepared and optimized for denitrification.

This is further corroborated by our observation of a linear accumulation of $^{15}$N labelled N$_2$ gas already from incubation start for our 15N incubation experiments as shown in Fig.1. The lack of a lag phase is frequently associated with 'active' denitrification (Bulow et al., 2010;Ward et al.,

2009). Conversely, denitrifiers in pure culture require a 24-48h reactivation period to recover from dormancy (Baumann et al., 1997;Baumann et al., 1996). There was no lag phase in any of our sponge tissue incubations, which strengthens our conclusion that the denitrifying community is active and optimized for the denitrification rates observed in our experiments. This again means

that the measured maximum denitrification rates are realistic and will occur *in situ* in situations where the sponge tissue becomes completely anoxic. This also suggests that the heterotrophic microflora in these sponges regularly find themselves in an anoxic or microoxic environment where it is beneficial to have the denitrifying apparatus turned on.

In the slurries of surface sediments from the Schulz Massive, *nir*K and *nir*S copy numbers were

comparable to those in the sponges (Table 2); however, in these samples we did not detect any labelled $N_2$ production within 48h of incubation. This would suggest that although a microbial community capable of denitrification is present in the surface sediments of the Schulz Bank, its activity was under detection limit. Low availability of reactive carbon in these Arctic sediments (Baumberger et al., 2016)  may be the reason for this lack of detectable denitrification activity, in

contrast to a high availability of reactive carbon within a living sponge. Our results indicate that in the Arctic deep sea, sponge grounds play a much more important role for nitrogen cycling and benthic-pelagic coupling than the surrounding sediment.

### 4.4 Sponge grounds as nutrient sinks

Considering denitrification rates of the three arctic and boreal sponge species investigated in this study as representative for boreal and arctic sponge grounds, we can calculate average nitrogen removal rates for boreal sponge grounds of 91 nmol N $cm^{-3}$ sponge $day^{-1}$ assuming all sponges are not pumping, and 42 nmol $cm^{-3}$ $day^{-1}$ when all sponges are pumping. For Arctic sponge grounds the rates will be 233 and 94 for non-pumping and pumping sponges, respectively. Based on our




own observations from several cruises, we conclude that masses of 10 kg m$^{-2}$ are common in boreal

sponge grounds, while smaller areas both in shelfs and fjords may even come up to densities of 30

kg m$^{-2}$. In other areas masses can be considerably lower and more patchy, e.g. 3.5 kg in the Traena

area, as reported by (Kutti et al., 2013). In the Arctic sponge grounds investigated in this study we

estimate the sponge biomass to be approximately 4 kg m$^{-2}$ based on our own observations.

These estimates reveal areal denitrification rates for the boreal sponge grounds of 0.764 mmol N

m$^{-2}$ sponge ground assuming non-pumping and still 0.354 mmol N m$^{-2}$ day$^{-1}$ assuming pumping

sponges. For Arctic sponge grounds the numbers are quite similar (sponge biomass is lower but

sponge denitrification rates are higher): 0.847 mmol N m$^{-2}$ day$^{-1}$ for pumping and 0.343 for non-

pumping sponges. These rates are well within the range - or, for the non-pumping situation, on the

upper end – of denitrification rates from continental shelf sediments, which are 0.1-1 mmol N m$^{-2}$

day$^{-1}$ (Middelburg et al., 1996;Seitzinger and Giblin, 1996).  For the most dense boreal sponge

grounds with sponge densities up to 30 kg m$^{-2}$, rates will be up to 2.3 mmol N m$^{-2}$ day$^{-1}$; 2-10 times

higher than in continental shelf sediments, and rather comparable to rates measured in coastal

sediments (e.g. (Asmala et al., 2017).

While our denitrification rates in sponges incubated under oxic conditions may reflect normal *in-situ* conditions for pumping sponges, our numbers on denitrification rates in sponges incubated

under anoxic conditions are theoretical extremes, since we do know little about the *in-situ* pumping

patterns of deep-sea sponges, and the environmental factors influencing them. Seawater nitrate

which fuels most of the denitrification under anoxic conditions enters the sponge through pumping.

The maximum denitrification rates in non-pumping sponges can therefore only be maintained until

the nitrate in the sponge pore water is used up. The length and frequency of these anoxic spells will

thus determine the variability of *in situ* sponge denitrification rates. Observations by Schläppy et

al. (2010b) showed non-pumping periods of sponges *in situ* of up to two hours, leading to complete

tissue anoxia, followed by several hours of high pumping activity. Sponges with dense tissue and

high loads of associated microbes (high-microbial abundance (HMA) sponges, such as most

sponges in our study) generally show slower volume pumping rates than sponges with low

microbial numbers and loose tissue structure (Weisz et al., 2008). Slow pumping rates lead to

reduced and heterogeneous oxygen concentrations in sponges (e.g. (Schläppy et al.,

2010b;Schläppy et al., 2007) while they still may supply sufficient nitrate from ambient seawater

to fuel denitrification. Even though our calculated areal denitrification rates of sponge grounds so

far only represent careful estimations, our study clearly shows that both boreal and arctic sponge

grounds can function as efficient nutrient sinks, especially when they reduce or stop pumping and

the tissue becomes anoxic. Environmental and anthropogenic stressors such as increased sediment

loads   (Bell et al., 2015) reduce pumping activity and increase anoxic conditions in sponges (Fang

et al., 2018; (Kutti et al., 2015;Tjensvoll et al., 2013), and thus stimulate nutrient removal through

denitrification. Increased future impact through multiple stressors affecting the ocean ecosystems

(Bopp et al., 2013) may thus lead to that deep-sea sponge grounds change their role in the marine

ecosystem from functioning mainly as nutrient sources to functioning mainly as nutrient sinks.

**Conclusions**

In this study we have shown that several sponge species actively remove the bioavailable nutrients

ammonium and nitrate from the marine ecosystem by denitrification and coupled nitrification-

denitrification, which challenges the common view of sponges as main nutrient providers through

mineralisation of organic matter. While variations in sponge remineralisation activity only

postpone the delivery of nutrients, denitrification inevitably removes these nutrients from the

marine ecosystem. The nitrogen cycle is not closed in the sponge grounds, the denitrified nitrogen, no matter of its origin, is no longer available as a nutrient and efficiently removed from the marine ecosystem. We further showed that the investigated sponges host an active community of denitrifyers which show highest denitrification rates under anoxic conditions. Anoxic conditions

occur when sponges are not pumping. Increased future impact of sponge grounds by anthropogenic stressors which reduce sponge pumping activity may thus lead to that deep-sea sponge grounds change their role in the marine ecosystem from nutrient sources to nutrient sinks.

*Data availability*. The data is available in the data publisher PANGAEA,

https://doi.pangaea.de/10.1594/PANGAEA.899821

*Author contribution*. F Hoffmann and C Rooks designed the study. C Rooks and J K-H Fang performed the sponge experiments. C Rooks and PT Mørkved performed the stable isotope analyses. C Rooks and R Zhao quantified the functional genes. C Rooks analyzed all the data. HT

Rapp organized the cruises, quantified sponge biomass at key sites and determined the sponge species. C. Rooks wrote the first draft of the manuscript, and all authors contributed substantially with writing and revision.  F Hoffmann supervised and coordinated the writing process, and finalized the manuscript.

*Competing Interests*. The authors declare that they have no conflict of interest.

*Acknowledgements.* This research has been performed in the scope of the SponGES project, which received funding from the European Union's Horizon 2020 research and innovation programme under grant agreement No. 679849. This document reflects only the authors' views and the



Executive Agency for Small and Medium-sized Enterprises (EASME) is not responsible for any

use that may be made of the information it contains.

This study also received funding from the Norwegian Research Council through the project

SedexSponge, Project Number: 225283.

5    Measurements of stable isotope samples was done at the Norwegian National Infrastructure

project FARLAB (Facility for advanced isotopic research and monitoring of weather, climate,

and biogeochemical cycling, Project Nr. 245907) at the University of Bergen, Norway.




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



**Table 1.** Nitrate sources for denitrification in the presence of dissolved oxygen. A large portion of nitrate removed by sponge denitrification in incubations with air-saturated seawater originates from sponge nitrification (coupled nitrification-denitrification).

| Sample | Location | Denitrification in oxic incubations nmol N cm$^{-3}$ sponge day$^{-1}$ | Nitrate from nitrification | Nitrate from seawater | % coupled nitrification-denitrification |
|---|---|---|---|---|---|
| *S. fortis* | Boreal | 0 | 0 | 0 | 0 |
| *G. atlantica* | Boreal | 40.96 | 29.52 | 11.44 | 72.06 |
| *G. barretti* | Boreal | 86.46 | 71.28 | 15.18 | 82.44 |
| *S. rhaphidiophora* | Arctic | 113.84 | 101.52 | 12.32 | 89.18 |
| *G. hentscheli* | Arctic | 86.73 | 72.2 | 14.53 | 83.25 |
| *G. parva* | Arctic | 82.09 | 55.15 | 26.94 | 67.18 |
| **Sediment** | Arctic | 0 | 0 | 0 | 0 |

**Table 2.** Abundance of the nitrite reductase genes *nir*S and *nir*K in sponge and sediment
10  samples. The nitrite reductase copy number is the sum of the mean number of *nir*S and *nir*K copies per cm$^{-3}$ of sponge tissue (n=3). *ND = not detectable.

| Sample | Location | *nir*S copy no. | *nir*K copy no. | Nitrite reductase copy no. |
|---|---|---|---|---|
| *S. fortis* | Boreal | ND | 2.19E+03 | 2.19E+03 |
| *G. atlantica* | Boreal | 2.67E+02 | 6.00E+07 | 6.00E+07 |
| *G. barretti* | Boreal | 7.04E+02 | 1.75E+06 | 1.75E+06 |
| *S. rhaphidiophora* | Arctic | 4.02E+02 | 2.39E+03 | 2.80E+03 |
| *G. hentscheli* | Arctic | 1.25E +03 | 1.82E+08 | 1.82E+08 |
| *G. parva* | Arctic | 3.81E+02 | 1.03E+09 | 1.03E+09 |
| *Sediment* | Arctic | ND | 2.77E+04 | 2.77E+04 |





## LIST OF FIGURE LEGENDS

**Fig. 1.** Production of $^{29}N_2$ (filled symbols) and $^{30}N_2$ (open symbols) as a function of time after the addition of $^{15}NO_3^-$ in incubations with **(A)** air-saturated (S.T.P, simulating pumping conditions) and **(B)** de-gassed site water (simulating non-pumping conditions) with tissue from *Geodia barretti*
(n=3 individuals). Data associated with an individual sponge is represented by a set of symbols. Linear regressions of $N_2$ production within the first 24 hours of the experiments were used to calculate denitrification rates. .

**Fig. 2.** Sponge species-specific rates of denitrification in incubations with de-gassed site water (anoxic conditions, black bars) and air-saturated site water (oxic conditions, grey bars) for 6 key
species from boreal and arctic sponge grounds. Statistically significant differences between denitrification rates in the presence and absence of dissolved oxygen are indicated by an asterisk for each species. Error bars indicate SE (n=3 individuals). Coupled nitrification-denitrification under oxic conditions is visualised with dark grey colour in the grey bars. Compare also Table 1.

**Fig. 3.** Mean species-specific denitrification rates in incubations with air-saturated site water (with
$O_2$, open circles) and de-gassed site water (without $O_2$, closed circles) as a function of nitrite reductase copy number. The nitrite reductase gene copy number is the sum of the mean number of *nir*S and *nir*K copies per $cm^{-3}$ of sponge tissue (n=3). A positive exponential relationship, based on the mean rates of denitrification and *nir* copy number in *S. rhaphidiophora (Sr)*, *G. barretti (Gb)*, *G. hentscheli (Gh)*, *G. phlegraei (Gp)*, is shown.



Figure 1A





Figure 1B



Figure 2





Figure 3

