# Peer review of "Deep-sea sponge grounds as nutrient sinks: Denitrification is common in boreo-arctic sponges"

_Biogeosciences, 2019_

## Referee Comment (RC1) · Anonymous Referee #1 · 6 May 2019

The paper presents an experimental study of dissimilatory nitrogen transformations in six cold-water sponge species with particular focus on their potential role as sinks for bioavailable nitrogen. Denitrification and anammox rates were quantified in oxic and anoxic incubations with N-15-labeled substrates, and nitrification rates were inferred from patterns of isotope pairing in N2. The process rates were supplemented with quantification of relevant functional genes.

The main result of the study is that the sponge microbiomes support substantial rates of denitrification under both oxic and anoxic conditions, and that denitrification under oxic conditions is driven to a large extent by nitrate produced endogenously by nitrification.

[Figure]

Upscaling of the rates indicates high nitrogen removal rates in sponge grounds.

Denitrification was previously demonstrated in a few sponge species, but this survey represents a substantial expansion of the small database, particularly for colder waters. It further contributes to the growing literature on nitrogen transformations in "exotic" environments such as marine snow, animal microbiomes, etc. Thus, it is an original and relevant study and well-suited for Biogeosciences. The study was generally well designed and the results are of good quality. The writing and presentation of results are generally clear. While some conclusions are justified others require further discussion and likely need to be moderated.

Major issues 1) Experiments were conducted with nitrate and ammonium added at at least 10 fold higher concentrations than in situ values (100 $\mu$M vs. 10 $\mu$M and 10 $\mu$M vs. $\leq$ 1 $\mu$M, respectively, i.e., 1000% above ambient, and not 90% as stated in the text p. 11 l. 6). This means that the measured rates must be treated as potential rates unless the authors can establish an argument for 0th-order kinetics for both denitrification and nitrification. In turn, this implies that the estimated sponge-ground rates may be vastly (10-fold) overestimated. This issue should be discussed and the conclusions modified accordingly. In the oxic experiments, denitrification rates could, in principle, be calculated using the classic isotope pairing calculations for sediment cores (D14 sensu Nielsen 1992), but then the incubations should have been performed without addition of unlabelled ammonium and with maintenance of steady state.

2) Nitrification-based denitrification rates are calculated from the accumulation of single labelled 29N2. Firstly, it is not entirely clear how these rates and relative contributions were calculated, and I suggest to include the essential equations in Methods. Secondly, the concept of water-based and nitrification-based denitrification was developed by Nielsen for sediment cores with steady state distributions of oxygen and nitrate (and it was challenged by Middelburg in L&O 41:1839). In the present study, oxygen was clearly not at steady state during the oxic incubations, and it also seems likely that new formed nitrate may have leaked from the sponge tissue thus gradually decreasing the

labelling of the ambient nitrate pool, and increasing 29N2 production from the ambient water. Moreover, the data presented in Fig. 1, for one of the six sponges, suggests that there is an issue with the mass balance of unlabelled N in the incubations. Thus, at the end of the anoxic incubations, excess 29N2 dominated over 30N2 in two of three incubations despite the stated ~90% labelling of the nitrate pool, and the accumulated 29N2, reaching up to ~23 $\mu$M, exceeds the amount of unlabelled nitrate initially available (10 $\mu$M in situ + 1 $\mu$M from the 99% 15N tracer). Also during the first 24 h, 29N2 production in the anoxic incubations seems higher than predicted by nitrate labelling in the absence of nitrification. Altogether, these uncertainties and discrepancies undermine the conclusion concerning the role of nitrification. Plots of excess 29N2 vs. excess 30N2 could potentially help the authors to evaluate and constrain some of these issues.

Specific comments 3, 8-12: The final statement is highly speculative and does not belong in an abstract.

4, 16-7: The statement about nif genes seems out of context.

6, 14: Science should never aim to show specific results but rather test hypotheses!

7, 11

9, 4-5: "Upper few centimetres" is vague – considering the negative result, the question is whether only the oxic surface layer was sampled.

9, 20: There was no "atmosphere" in the vials? However, incubation with a helium/oxygen headspace would have kept the incubations oxic throughout.

10, 7-8: This seems a very shaky assumption. Respiration rates must vary with species, temperature, and trophic state.

10, 18-9: Some oxygen is likely introduced during transfer – did you test the water in the Exetainers?

11, 6: The values are $\geq$1000% above ambient.

11, 12: According to 7, 11 the in situ temperature was below 0 °C! How would the higher incubation temperature affect the rates?

12, 18: The accumulations in Fig. 1 look only approximately linear – which test gave p < 0.05? Did the same apply to the linearity of the anoxic rates (13, 4)?

13, 15: Please specify the equations used here (see major issue #2).

16, 3-5: The opening of the Results is very confusing with the first two sentences referring to two different treatments. Delete the first sentence.

16, 22-3: The sediment experiment has little value. The origin of the sediment is unclear, and it does not seem representative of Arctic sediments.

18, 5: See 6, 14.

18, 18-9: Metabolisms in sponges or what? Please clarify/reference.

18, 20-5: The presence of denitrification genes and isolation of denitrifiers cannot prove "the presence of denitrification activity".

20, 11: How would the "pulse of organic matter in the water column" (where in the water column?) affect potential denitrification in the sponges' tissue?

21, 16: "proves" is an overstatement.

22, 1-2: It is not the in situ concentration but the 10 $\mu$M ammonium added, that is of relevance here.

22, 13-5: Please provide a reference for the single copies.

22, 16-20: The curve in Fig. 3 does not look like an exponential function. It there statistical support for this relationship?

22, 20: What is meant by "optimized"?

23, 9-10: With 6 orders of magnitude variation, this is not very telling.

23, 19 on: The calculations of sponge ground rates need explanation, but see Major issue #1. Furthermore, it seems that results of population density surveys are presented here for the first time. If this is the case, the methods and results should be specified i the appropriate sections. Otherwise, a reference should be included.

24, 24: What was the frequency of non-pumping?

25, 11-2: Is this a short-term or permanent effect? Would reduced pumping rates/increased anoxia not result in reduced growth, reduced biomass, and thereby reduced nitrogen removal in a longer perspective? The system effect of the stressors seems speculative.

Table 1: The number of significant digits should be adjusted.

Fig. 1: Different triangles are used for 29N2 and 30N2.

―――――――――――――――――――

---

## Referee Comment (RC2) · Anonymous Referee #2 · 8 May 2019

Biogeosciences review

General Comments:
This is an interesting and well-planned study. It is nicely focused and well-suited to address the question outlined by the authors. I do not see any major flaw with the experimental design or the interpretation, however, I think there are several places where some more clarity and/or improved organization would be helpful to the readers. My "major" comments are that the authors could setup arctic vs boral comparison a little more purposefully and clearly in the introduction. Also, a map of the sampling are would be really helpful as I (and I think most people) do not have a good image of this region in my head, it could be a supplemental file if need be.
There are also a number of typos and grammatical errors that need to be addressed. Aside from those, I have also outlined below some minor comments that may help improve clarity and a few places where some more context or broader discussion is warranted. Overall, I enjoyed reading this manuscript and the methods, statistics, and interpretation are sound.

Specific Comments:
Pg 4
L8: I would think that each compound should be defined first with the name and formula in parentheses, then you can use the formula after that
L10-13: This sentence and really the whole first pp is on one hand a logical introduction, yet it still leaves me with the thought "what is the point of this pp"? Can the authors make it a more cohesive and setup the transition to N fixation (in the next pp) a little better? (and in the second pp it starts with N fix then goes back to DIN)
L22-26: This last sentence in the paragraph could be clearer. I follow the first half ok, but the second half starting on L24 is not clear to me.
L21: This goes back to a topic discussed in the first pp so I think some more context is needed here because it is a sudden transition to an earlier topic. Also what "observed variations" do the authors refer to here on L22?
L22: Maybe use "DIN" instead of "nutrient" to be more specific
L22: Instead of "a combination of variations" maybe just use "variability"

Pg 6
L4: This pp might be a good place to clearly introduce arctic vs boreal

Pg 5
L4: This might be a good place to briefly discuss the dependence of nitrification on oxygen, which is not really stated anywhere upfront but that information is relevant think.
L14: There is no anaerobic process quantified in Fiore et al. 2013

Pg7
L21: average biomass for boreal sponges is given, is this true for arctic sponges too?

Pg3

L2 and 6: what is meant by "key" here, why were these species chosen?

L9-10: says that sediment was not collected here, so it seems like something about sediment should be mentioned in the paragraph before this one.

L17-22: I can tell the authors tried to make the sampling and setup of experiments clear but I am still a little confused. For one, it would be helpful to describe the shape of the sponges, presumably  massive/round? Second, why were these cut into three pieces- for each isotope tracer, is that right?

Pg 9

L17: This setup was on the ship yes? Might be good to remind the reader of that

L18: Why  only  sand filter for the boreal specimens?

Pg11 L13: Can the authors say how many samples were sampled at each time point rather than "a selection" – or am I missing something here?

Pg13 L12: This seems redundant because of pg 12, which would be ok, but it makes it a little confusing

Pg13 L23: Is it possible to give a little more guidance on the calculation of nitrification derived nitrate? It would be helpful for the reader and worth the word count since this becomes an important piece of information in the discussion.

Pg 14: Out of curiosity, why was amoA not quantified? Since the authors are interested in nitrification rates as well

Pg 17 L3 – I suggest adding some of these calculations (even brief) in a supplemental document

Pg18 L23: "proving" is a strong word… maybe "demonstrating" works better? (when possible in other instances for the use of "prove" is there another word that could work?)

Pg18 L24: Fiore et al. 2010 is a review paper so I don't know that that fits in here with this sentence

Pg19

L3-9: The information in this pp is fine, but I don't really see the point of the pp. It could use some language to tie it into the paper more.

L11: This sentence is awkward, try to reword without using "being"- just using "As denitrification is an…" would be much more straightforward.

L15-16: certainly interesting, but this sentence also leaves me wondering, what is the point? It would be nice to have it tied in more clearly for the reader.

Pg20

L1-6: It would be helpful to have more context here on any work that has been done to measure pumping activity – has anyone done this? Do we have any idea of these deep sea sponges behave the same as others? The authors do get to this type of info later in the discussion but this pp seems lacking a bit as is.

L7-8: The use of "(anaerobic)" and "(some)" is confusing to me

Pg21 L16-17: The first point here about oxygen in the specimens here vs explants is confusing as to what the point is. I think I get what the authors want to say but it is not all that clear  here and could be tied in better.

Pg22 L20: "prepared and optimized" – I get hung up on "optimized" here, is there better way to say this? I think the authors mean the community is well suited to this environment or adapted to this environment, but optimized sounds odd to me

Pg23 L8: instead of "apparatus turned on" maybe say "genes expressed"

Pg24 L14: would be helpful to give some of this info briefly (sediment denitrification rates)

Pg25 L19-22: seems more to the point here to contrast with nitrification studies showing release of nitrate, rather than discussing "nutrients" as a whole which is vague

Pg26 L4-5: This sentence is not tied in well, so it reads a bit awkward

Pg26 L6-7: This last sentence is a bit awkward and unclear at the end.

Technical Comments:

L10 abstract: "thus lead to that"

pg 5 L21: "This opens for"

Pg 7  L15: check reference format

Pg 15 L 6: "Standard of qPCR"

Pg19 L21: parentheses for references- it looks like it should be: (e.g., Wilson….).

Pg22 L4: "origin" should be "originate" I think

Pg24 L10-11: are the "-" supposed to be there?

Pg24 L18: "since we do know" – maybe just remove "do"

Pg25 L5 and 12: extra parenthesis

---

## Referee Comment (RC3) · Anonymous Referee #3 · 6 Jun 2019

**Review Rooks et al 2019, 135**

**General comments**

This manuscript provides novel information on potential (de)nitrification and anammox rates combined with genomics in 6 abundant cold-water sponges, from which 5 have not been analyzed previously. The data show that denitrification is a common process in deep-sea sponges and is relevant for understanding the role of sponges in nutrient cycling. The study seems well planned and conducted.

My main concern is that the potential denitrification rates measured in this study in tissue sections are upscaled to whole sponges and ecosystem level and are even used for future predictions under anthropogenic stress. The rates here should be treated as maximum or potential rates, since they were conducted with 10 times ambient concentrations, on small tissue sections in closed exetainers, under no oxygen and decreasing oxygen concentrations.

My second major comment is that the MS is focused on and biased towards denitrification, with limited attention for other nitrogen transforming processes, such as nitrification, anammox and perhaps DNRA. I suggest to present all labelling incubations, carefully evaluate the results and present and discuss in a more balanced overview of the different nitrogen transforming processes and also include all data in the published Pangea dataset.

**Specific comments**

- Title: The term "nutrient sink" in the title is confusing and questionable
- P3, L3-11: this is too speculative, see major comment 1.
- P4, L16-19: This sentence doesn't really fit and perhaps the whole part of N fixation can be moved to the discussion, since it disrupts the introduction on DIN release.
- P7, L20: Add some of the relevant characteristics for this site.
- P2.2 and p2.3: A table or flow chart with the experimental incubations would be very useful.
- P10, L:16: On the previous page it is mentioned that all incubations were done with water sampled from the deep, but here surface site water is mentioned for anoxic incubations. This needs to be clarified.
- P11, L4: Can you be sure that $15-NO_3$ is reduced to $15-NO_2$, the preferred substrate for anammox? Please elaborate
- P 11, L6-7: This is 1000% above ambient concentrations, and ambient concentrations from which site, arctic or boreal grounds or both?
- P11, L12: This is not *in situ* temperature for the Arctic species, the temperature increase might increase your potential rates.
- Paragraph 2.3.2: There were no oxic sediment incubations?
- Paragraph 2.3.3: Including the calculations is informative for the reader.
- P12,L22-P13, L1: The published dataset contains individuals with tissue degradation
- P13, L:11-13: This needs more explanation. I guess you mean no $29-N_2$ was detected in the anoxic incubations? What about $29-N_2$ and $30-N_2$ production in oxic incubations with

labeled ammonium? Some production can be expected from coupled nitrification and denitrification?

- Paragraph 2.3.4: Also here, the equations would be useful. And following my comment above, can you estimate coupled nitrification-denitrification from your oxic incubations with $^{15}N-NH_4$?
- Paragraph 2.4: In the results and discussion is mentioned that the sponges were also screened for anammox and N2 fixation functional genes. The screening and description of the functional genes should be described here. Was there also screening for other genes relevant for the nitrogen cycle (e.g. nitrification, DNRA)?
- P16, L1-6: Add graphs or tables with the results under oxic and anoxic conditions.
- P16,L22-23: What about unlabeled $N_2$ production?
- P18, L5: I would remove "nutrient removal"
- P18, L22: Denitrification has not been directly shown in Fiore et al. 2013, but is given as a potential pathway, together with anammox or DNRA, to explain net consumption of nitrate in some of the sponges.
- P20, L6: I won't state that results are representative for normal conditions, but state that these conditions are not atypical (or something similar).
- P20, L9-L15: I would expect year-round higher (dissolved) organic matter concentrations at the Boreal compared to Arctic grounds. Another explanation might be related to the higher incubation temperature (if it was 6°C) compared to the *in situ* temperature.
- Paragraph 4.2: The relevance of your denitrification rates in view of other nitrogen transforming processes should be discussed in a balanced way (see major comment 2). This paragraph gives the impression that denitrification is more important than nitrification in sponges, even though the majority of sponge studies reveal that sponges are net sources of nitrate, with denitrification being only a fraction of nitrification. Also the possibility of DNRA as competitive process for denitrification should be discussed somewhere.
- P21, L11-14: What is so different between explants and tissue sections? The tissue sections will also dependent on diffusion? There are more differences between Hoffmann et al. 2009, i.e. in your study you added $NH_4$, which will stimulate nitrification, while in Hoffmann et al. 2009, no $NH_4$ was added. You could discuss the reliability of nitrification measurements.
- P21, L21-25: "May be higher" should be "are likely higher" and the reported rates are really at the low end of other reported rates.
- P22, L1, yes, but you added 10 µM (unlabeled) $NH_4$, which can result in 10 µM (unlabeled) $NO_3$ in oxic conditions.
- P22,L6,9: These last two sentences are not connected to the rest of the paragraph.
- P22, L16-21: I won't use optimized, I guess you want to say there is an active denitrifying community. Perhaps add some statistics to the relationship, this is a nice result.
- P23, L4-6: I disagree that they are realistic, see major comment one.
- P23, 15-17: Are there reported denitrification measurements of Arctic sediments? What about the other nitrogen transforming processes? A comparison to literature should be added with to this statement.

- Paragraph 4.4: The results are not representative for a natural situation, but rather show a potential, so I would be extremely cautious to upscale these numbers and refer to these sponges as efficient nutrient sinks (see major comment 1).
- P24, L6-10: The calculations and conversion factors going from volume to surface integrated measurements are lacking (but see major comment 1).
- P25, L13-15: Combined anthropogenic stressors can also lead to changes in nutrient and organic matter availability which might affect microbial composition and biogeochemical processes. It is too speculative to state that sponges will become nutrient sinks in the future if they reduce pumping.
- P26, L10: Please expand the dataset in Pangaea with all incubation data and results.
- P35, L2: STP is used for the first time, does it stand for standard temperature and pressure?
- Figure 1 and figure 2 should be swapped, based on their reference order in the text.
- The order of references in the citations and full references need to be checked.

---

## Author Comment (AC1) · 30 Sep 2019

Dear editor, dear reviewers,

Sorry for the delay and thank you for your patience. Please find our replies to all reviewers' comments uploaded. Here we summarize the most important changes in the manuscript:

1. We realized that the incubation experiments were not clearly described in the method section, in particular it was not clear which kind of 15N species was added to which experiment. This caused some fundamental misunderstanding about the

method for Ref 1 and 3. This is now corrected and explained. 2. Challenged by some critical comments by Ref 1 and 3 about calculation of denitrification rates, the first author CR re-calculated all the rates from scratch, going back to integration of peak areas and data plotting. This resulted in a change of some of the final denitrification rates, which however did not affect the main story of the manuscript. 3. By going through the entire raw data set again, CR discovered a column mix-up between 29N and 30N data, which has lead to the surprisingly high rates for coupled nitrification-denitrification in our previous manuscript version. The new rates make much more sense and make a better fit with previously reported values. 4. We updated figures and tables accordingly, an updated data set was sent to PANGAEA.

We are aware that in an ideal world, we should have discovered these errors already before the first submission. However, this also shows that the peer review system works, and we are grateful to the critical reviewers who helped to improve the manuscript.

Finally, we are aware that there are some mistakes in the format of the in-text citations and the reference list. Reference format is generated automatically through EndNote each time we open the document. We will do the final formatting manually when the manuscript is accepted and no references have to be removed or added.

I also would like to send the revised data set (the PANGAEA publication is not publicly available yet) but do not know how to do this (it is in excel format). I submitted the cleaned version of the manuscript (no tracked changes) as a figure as I did not find any other options to post it. I submit the revised manuscript including tracked changes as supplement file.

Please also note the supplement to this comment:
https://www.biogeosciences-discuss.net/bg-2019-135/bg-2019-135-AC1-supplement.pdf

[Figure]

[Figure]

**Fig. 1.**

[Figure]

**Fig. 2.**

[Figure]

**Fig. 3.**

[Figure]

[Figure]

**Fig. 4.**

[revised manuscript text omitted]

Denitrification in sponges thus opens for an alternative role of sponges as nutrient scavengers, and an alternative explanation for the  variations in DIN release as described

above: an interplay of both, variability in remineralisation rates associated with food availability **and** direct consumption of endogenous and ambient nutrients by microbial processes in sponges. Sponges can even perform competing nitrogen cycling processes such as nitrification and denitrification simultaneously(Hoffmann et al.,

5    2009a), where the rates of the different processes determine whether the sponge acts 
[revised manuscript text omitted]

| *G. hentscheli* | Arctic | 1.25E +03 | 1.82E+08 | 1.82E+08 |
| *Sediment* | Arctic | ND | 2.77E+04 | 2.77E+04 |

---

## Author Comment (AC2) · 30 Sep 2019

Referee #1 Reply

Major issues
1) Experiments were conducted with nitrate and ammonium added at at
least 10 fold higher concentrations than in situ values (100 μM vs. 10 μM and 10 μM vs.
1 μM, respectively, i.e., 1000% above ambient, and not 90% as stated in the text p.
11 l. 6).
*We rephrased the text to avoid misunderstanding (p 12 line 8)*

This means that the measured rates must be treated as potential rates unless
the authors can establish an argument for 0th-order kinetics for both denitrification
and nitrification. In turn, this implies that the estimated sponge-ground rates may be
vastly (10-fold) overestimated. This issue should be discussed and the conclusions
modified accordingly.
*We agree that these are potential rates. We make this now more clear in the discussion, beginning of chapter 4.4.*
*$^{15}N$ incubations were based on standard methods (Dalsgaard et al., 2003 and Hannig et al., 2007) with minor modifications as per Hoffmann et al. 2009. These methods allow us to estimate the rates at ambient $NO_3$ concentrations based on 15N incubations. They are not the rates measured directly by the 15N labelled N2 production. This is clarified in Section 2.3. Although concentrations of labelled $^{15}N^-$ exceeded background $^{14}N$ 10-fold, potential denitrification rates were well within the range of those previously reported for cold and warm water sponges, where $^{15}N$ amendments were more reflective of ambient $NO_3^-$ concentrations (Hoffmann et al., 2009;Schläppy et al 2010a.*
In the oxic experiments, denitrification rates could, in principle,
be calculated using the classic isotope pairing calculations for sediment cores (D14
sensu Nielsen 1992), but then the incubations should have been performed without
addition of unlabelled ammonium and with maintenance of steady state
*Labelled ammonium was added to the annamox incubations. No ammonium was added to the denitrification experiment. We now discovered that this was not clearly explained in the method section, which have led to confusion. The text is now corrected.*
*Calculations of coupled nitrification-denitrification were based on the approach of Hoffmann et al., 2009, where a similar experimental set-up was employed. The potential rates of denitrification and proportion of coupled nitrification-denitrification are comparable for Geodia barretti – 92 nmol N cm-3 sponge day-1 ; 26% coupled nitrification-denitrification (Hoffmann et al., 2009) relative to 96 nmol N cm-3 sponge day-1; 16% coupled nitrification-denitrification (this study).*

2) Nitrification-based denitrification rates are calculated from the accumulation of single
labelled 29N2. Firstly, it is not entirely clear how these rates and relative contributions
were calculated, and I suggest to include the essential equations in Methods.

*The abundance of $^{28}N_2$, $^{29}N_2$ and $^{30}N_2$ were analysed from gas samples using a continuous flow isotope ratio mass spectrometer (CF/IRMS). Calibrations were achieved by injecting lab air and additional in house reference gas samples. Denitrification rates were calculated from the production of $^{15}N$ isotopes (see below) according to the method described by Nielsen (1992).*

*The rate of denitrification was measured from $^{15}N$ isotope production (equations 1 and 2). $D_{14}$ and $D_{15}$ represent denitrification of labelled $^{15}NO_3^-$ and $^{14}NO_3^-$. p ($^{14}N^{15}N$) and p ($^{15}N^{15}N$) are the production rates of the 2 labelled $N_2$ species $^{14}N^{15}N$ and $^{15}N^{15}N$ (Rysgaard et al.1995). $D_{15}$*

*is indicative of denitrification of labelled $^{15}NO_3^-$ and $D_{14}$ represents in situ denitrification of $^{14}NO_3^-$.*

$$D_{15} = p\,(^{14}N\,^{15}N) + 2p\,(^{15}N\,^{15}N) \tag{1}$$

$$D_{14} = \frac{p\,(^{14}N\,^{15}N)}{2p\,(^{15}N\,^{15}N)}\,D_{15} \tag{2}$$

*To estimate denitrification of $NO_3^-$ from ambient sea water ($D_w$), in terms of $D_{14}$, the following calculation was applied (equation 3):*

$$D_w = D_{15}\,[^{14}NO_3]_w / [^{15}NO_3]_w \tag{3}$$

*where $[^{14}NO_3]_w$ and $[^{15}NO_3]_w$ represent the concentration of unlabelled and labelled $NO_3^-$ in ambient seawater.*

*In situ coupled denitrification ($D_n$), in terms of $D_{14}$, was calculated using equation 4 (see below).*

$$D_n = D_{14} - D_w \tag{4}$$

*These equations are now given in the method section, page 15.*

Secondly,
the concept of water-based and nitrification-based denitrification was developed
by Nielsen for sediment cores with steady state distributions of oxygen and nitrate (and
it was challenged by Middelburg in L&O 41:1839). In the present study, oxygen was
clearly not at steady state during the oxic incubations, and it also seems likely that new
formed nitrate may have leaked from the sponge tissue thus gradually decreasing the
C2 labelling of the ambient nitrate pool, and increasing 29N2 production from the ambient
water. Moreover, the data presented in Fig. 1, for one of the six sponges, suggests
that there is an issue with the mass balance of unlabelled N in the incubations. Thus,
at the end of the anoxic incubations, excess 29N2 dominated over 30N2 in two of three
incubations despite the stated _90% labelling of the nitrate pool, and the accumulated
29N2, reaching up to _23 μM, exceeds the amount of unlabelled nitrate initially available
(10 μM in situ + 1 μM from the 99% 15N tracer). Also during the first 24 h, 29N2
production in the anoxic incubations seems higher than predicted by nitrate labelling
in the absence of nitrification. Altogether, these uncertainties and discrepancies undermine
the conclusion concerning the role of nitrification. Plots of excess 29N2 vs.
excess 30N2 could potentially help the authors to evaluate and constrain some of these
issues.

*Ambient seawater was filtered to remove water column bacteria and or phytoplankton, thus reducing the potential for background nitrification. Although conditions were not at steady state, previous application of this method was considered suitable for dissected sponge explants (Hoffmann et al., 2009). The data has been re-analysed as suggested and errors in the calculation have been corrected. Issues with mass balance and $^{29}N_2$ production have now been resolved.*

Specific comments 3, 8-12: The final statement is highly speculative and does not belong in an abstract.

*We agree that this statement sounds provocative, but we still consider this a valid interpretation of our data, see justification below for comment on 25,11*

4, 16-7: The statement about nif genes seems out of context. *Agree, removed*

6, 14: Science should never aim to show specific results but rather test hypotheses! *Rephrased 7, 11*

9, 4-5: "Upper few centimetres" is vague – considering the negative result, the question is whether only the oxic surface layer was sampled.

*As the reviewer pointed out, the "upper few centimetres" sediments are very likely oxic (our microsensor measurements indicate that oxygen can penetrate to ~75 cm below seafloor). However, denitrification (at also other anaerobic processes) is most active in the upper most sediments, due to the widespread of bioturbation and bioirrigation in marine surface sediments (e.g. see the recent paper regarding sulfate reduction rates in the Aarhus Bay sediments (Andrew Dale et al, 2019, GCA)). Therefore, measurements made using the most surface sediments are believed to represent the majority of denitrification activities in a marine sediment column. (Of course, this is supported by our reaction-transport model!)*

9, 20: There was no "atmosphere" in the vials? However, incubation with a helium/ oxygen headspace would have kept the incubations oxic throughout.

*No atmosphere in the exetainers as described on p11. We followed a standard protocol here.*

10, 7-8: This seems a very shaky assumption. Respiration rates must vary with species, temperature, and trophic state.

*They certainly do, but we do not have these details for all investigated species and needed to make the best possible estimation.*

10, 18-9: Some oxygen is likely introduced during transfer – did you test the water in the Exetainers?

*To verify the absence of oxygen in the de-gassed water, an anaerob strip test (colour change from pink to white under anaerobic conditions; Sigma Aldrich) was performed prior to transfer into 12mL exetainers. The caps were then replaced and the gas tight vials were carefully sealed to exclude any air bubbles. An anaerob strip was added to control exetainers (seawater only) to verify the absence of oxygen in anaerobic incubations. See p 11, line 19*

11, 6: The values are _1000% above ambient.

10 times above, corrected.

11, 12: According to 7, 11 the in situ temperature was below 0 _C! How would the higher incubation temperature affect the rates?

*Lab experiments can never perfectly mimmick in situ conditions, and in our case, there were no cool room available at 0 C. We are aware that this may have led to over-estimation of the Arctic rates and made a comment in the discussion.*

12, 18: The accumulations in Fig. 1 look only approximately linear – which test gave p < 0.05? Did the same apply to the linearity of the anoxic rates (13, 4)? *Figures modified after re-calculation of rates*

13, 15: Please specify the equations used here (see major issue #2).

*See above, our reply to major issue #2.*

16, 3-5: The opening of the Results is very confusing with the first two sentences referring to two different treatments. Delete the first sentence.

*The first two sentences explain why our results show the absence of anammox, they refer to the same treatment. We rephrased for more clarity.*

16, 22-3: The sediment experiment has little value. The origin of the sediment is unclear, and it does not seem representative of Arctic sediments.

*The origin of the sediment samples (next to Arctic sponge ground at Schulz Massiv) is clearly stated in chapter 2.2. The sediment itself is of pelagic origin (ultimately from the primary production in the surface ocean rather than terrestrial origin), this is obvious from the geographic position of the sampling site and does not need to be mentioned.*

18, 5: See 6, 14.

*Rephrased*

18, 18-9: Metabolisms in sponges or what? Please clarify/reference.

*Rephrased for clarification*

18, 20-5: The presence of denitrification genes and isolation of denitrifiers cannot prove "the presence of denitrification activity".

*Rephrased*

20, 11: How would the "pulse of organic matter in the water column" (where in the water column?) affect potential denitrification in the sponges' tissue?

*Section rephrased*

21, 16: "proves" is an overstatement.

*Changed to "shows"*

22, 1-2: It is not the in situ concentration but the 10 µM ammonium added, that is of relevance here.

*Disagree. Ammonium was only added to the annamox experiments, not to the denitrification experiments. So we have a point here.*

22, 13-5: Please provide a reference for the single copies.

*This is text book knowledge, we do not see a need to provide a reference*

22, 16-20: The curve in Fig. 3 does not look like an exponential function. It there statistical support for this relationship?

*The figure changed after re-calculating the data. The point is that there is a clear positive correlation between anaerobic denitrification rates and nir genes for most species, we made this more clear now.*

22, 20: What is meant by "optimized"?

*Rephrased*

23, 9-10: With 6 orders of magnitude variation, this is not very telling.

*Our point is that there were not less nir genes in the sediment than in some of the sponges, but no denitrification.*

23, 19 on: The calculations of sponge ground rates need explanation, but see Major issue #1.

*See text added and rephrased at beginning of chapter 4.4*

Furthermore, it seems that results of population density surveys are presented here for the first time. If this is the case, the methods and results should be specified i the appropriate sections. Otherwise, a reference should be included.

*This is described in the methods, p 8 l 21-22. We rephrased also in section 4.4 to make clear that these are careful estimates.*

24, 24: What was the frequency of non-pumping?

*Not possible to say something in general because this varies between species, environmental conditions etc. In the particular study quoted here, there was one non-pumping period of 1-2*

*hours during the experiments of 12-20 hours, this can easily be looked up in the cited reference.*

25, 11-2: Is this a short-term or permanent effect? Would reduced pumping rates/increased anoxia not result in reduced growth, reduced biomass, and thereby reduced nitrogen removal in a longer perspective?

*Reduced growth would first lead to reduced remineralisation activity – so it would first of all weaken the classical DIN source function of the sponge. We know for sure that reduced pumping leads to reduced oxygen in sponge tissue, but we do not know if reduced pumping leads to reduced growth, so there is no point speculating about it here.*

The system effect of the stressors seems speculative.

*We rephrased to make clear that this is not speculation, but a potential scenario based on valid data interpretation.*

Table 1: The number of significant digits should be adjusted.

*Adjusted to what? There are two digits per value, what is wrong?*

Fig. 1: Different triangles are used for 29N2 and 30N2.

*Resolved*

---

## Author Comment (AC3) · 30 Sep 2019

Ref #2

Answers

Biogeosciences review
General Comments:
This is an interesting and well-planned study. It is nicely focused and well-suited to address the question outlined by the authors. I do not see any major flaw with the experimental design or the interpretation, however, I think there are several places where some more clarity and/or improved organization would be helpful to the readers.

My "major" comments are that the
authors could setup arctic vs boral comparison a little more purposefully and clearly in the introduction.
*Done, p 6.*
Also, a map of the sampling are would be really helpful as I (and I think most people) do not have a good image of this region in my head, it could be a supplemental file if need be.
*We now provide a simple map, which we suggest to make available as supplementary information (Fig suppl). Geographical position of the sampling sites can also be visualized through the map tool in the Pangaea database.*

There are also a number of typos and grammatical errors that need to be addressed. Aside from those, I have also outlined below some minor comments that may help improve clarity and a few places where some more context or broader discussion is warranted. Overall, I enjoyed reading this manuscript and the methods, statistics, and interpretation are sound.

Specific Comments:
Pg 4
L8: I would think that each compound should be defined first with the name and formula in parentheses, then you can use the formula after that
*OK, corrected*
L10-13: This sentence and really the whole first pp is on one hand a logical introduction, yet it still leaves me with the thought "what is the point of this pp"? Can the authors make it a more cohesive and setup the transition to N fixation (in the next pp) a little better? (and in the second pp it starts with N fix then goes back to DIN)
*We rephrased according to suggestion*
L22-26: This last sentence in the paragraph could be clearer. I follow the first half ok, but the second half starting on L24 is not clear to me.
*removed*

P5
L21: This goes back to a topic discussed in the first pp so I think some more context is needed here because it is a sudden transition to an earlier topic. Also what "observed variations" do the authors refer to here on L22?

L22: Maybe use "DIN" instead of "nutrient" to be more specific
L22: Instead of "a combination of variations" maybe just use "variability"
*Entire sentence (L21-24) rephrased*

P6
L4: This pp might be a good place to clearly introduce arctic vs boreal
*Done*

Pg5
L4: This might be a good place to briefly discuss the dependence of nitrification on oxygen, which is not really stated anywhere upfront but that information is relevant think.
*The information is given in the same sentence but we rephrased to make it more clear*

L14: There is no anaerobic process quantified in Fiore et al. 2013
*OK, reference removed here*

Pg7
L21: average biomass for boreal sponges is given, is this true for arctic sponges too?
*Yes, estimates for average biomass for both boreal and arctic sponge grounds is given in the discussion (chapter 4.4) and in the data sheet published in Pangaea*

Pg3 – *you mean Pg8?*
L2 and 6: what is meant by "key" here, why were these species chosen? *We chose species that are typical and representative for this type of sponge ground, see added text about sponge ground characterisation*
L9-10: says that sediment was not collected here, so it seems like something about sediment should be mentioned in the paragraph before this one.
*Moved this sentence to the section on sediment sampling*
L17-22: I can tell the authors tried to make the sampling and setup of experiments clear but I am still a little confused. For one, it would be helpful to describe the shape of the sponges, presumably massive/round? Second, why were these cut into three pieces- for each isotope tracer, is that right? *Details on sponge shape added in text. The sponges were cut into three sections to aid dissection. To ensure that we used only the choanosomal portion of the tissue, the most practical way to dissect this from a large individual was to cut the sponge into three pieces. Three whole sponges (n=3) were collected for each species. The dissected tissue from a single sponge represents one replicate. Also these details were now added to the text.*

Pg 9
L17: This setup was on the ship yes? Might be good to remind the reader of that
*On the ship for arctic species, in the lab for boreal species – clearly stated in line 13-15*
L18: Why only sand filter for the boreal specimens?
*It was the only option available in that lab facility.*
Pg11 L13: Can the authors say how many samples were sampled at each time point rather than "a selection" – or am I missing something here?
*Corrected: 3 samples per species (one for each replicate specimen)*
Pg13 L12: This seems redundant because of pg 12, which would be ok, but it makes it a little Confusing

*As we did not detect any anammox, we could not calculate the rates or contribution of anammox to total N2 production according to this method. Is this what the referee means by redundan?.*

Pg13 L23: Is it possible to give a little more guidance on the calculation of nitrification derived nitrate? It would be helpful for the reader and worth the word count since this becomes an important piece of information in the discussion.
*Section is now extended including equations*

Pg 14: Out of curiosity, why was amoA not quantified? Since the authors are interested in nitrification rates as well
*This work focusses on denitrification. Coupled nitrification/denitrification was of interest in this context, but it was never our intention to quantify total nitrification rates or the genes involved. Nitrification in sponges is well explored already, while denitrification is not.*

Pg 17 L3 – I suggest adding some of these calculations (even brief) in a supplemental document
*Done*

Pg18 L23: "proving" is a strong word… maybe "demonstrating" works better? (when possible in
other instances for the use of "prove" is there another word that could work?)
*Rephrased*

Pg18 L24: Fiore et al. 2010 is a review paper so I don't know that that fits in here with this sentence
*OK we took it out*

Pg19
L3-9: The information in this pp is fine, but I don't really see the point of the pp. It could use some language to tie it into the paper more.
*The aim of this paragraph is to explain why we were not able to reproduce the (very low) anammox rates that we had quantified in one of the species previously. If the reviewer and the editor advice to delete this paragraph, we will do so.*

L11: This sentence is awkward, try to reword without using "being"- just using "As denitrification is an…" would be much more straightforward.
*OK, rephrased*

L15-16: certainly interesting, but this sentence also leaves me wondering, what is the point? It would be nice to have it tied in more clearly for the reader.
*The point is that an aerobic process and an anaerobic process happen at the same time. Even though this has been described before, it is unusual since aerobic processes usually take over as soon as oxygen is present. We think we have a point here?*

Pg20
L1-6: It would be helpful to have more context here on any work that has been done to measure pumping activity – has anyone done this? Do we have any idea of these deep sea

sponges behave the same as others? The authors do get to this type of info later in the discussion but this pp seems lacking a bit as is.
*We think that the references we give here and the information we provide later is sufficient.*
L7-8: The use of "(anaerobic)" and "(some)" is confusing to me
*Agree, clarified*

Pg21 L16-17: The first point here about oxygen in the specimens here vs explants is confusing as to what the point is. I think I get what the authors want to say but it is not all that clear here and could be tied in better.
*Section entirely rephrased, as re-calculation of the data gave a much better fit with literature values, so there is no need any more to explain differences.*

Pg22 L20: "prepared and optimized" – I get hung up on "optimized" here, is there better way to say this? I think the authors mean the community is well suited to this environment or adapted to this environment, but optimized sounds odd to me
*Agree, rephrased*

Pg23 L8: instead of "apparatus turned on" maybe say "genes expressed"
Agree, rephrased

Pg24 L14: would be helpful to give some of this info briefly (sediment denitrification rates)
*I think there is no need to go into detail, this information can be looked up in the cited references.*

Pg25 L19-22: seems more to the point here to contrast with nitrification studies showing release of nitrate, rather than discussing "nutrients" as a whole which is vague
*No, it is the contrast with both nitrification studies showing release of nitrate, and mineralisation studies showing release of ammonium. Good point, we made this more clear now.*

Pg26 L4-5: This sentence is not tied in well, so it reads a bit awkward
*Rephrased*

Pg26 L6-7: This last sentence is a bit awkward and unclear at the end.
*Rephrased*

Technical Comments:
L10 abstract: "thus lead to that"
*We were not able to find this sentence*
pg 5 L21: "This opens for"
*Rephrased*
Pg 7 L15: check reference format
*Reference format is generated automatically in EndNote. We will do the formatting when the manuscript is accepted and no references have to be removed or added.*
Pg 15 L 6: "Standard of qPCR"
Rephrased to "qPCR standard".
Pg19 L21: parentheses for references- it looks like it should be: (e.g., Wilson….).

*Reference format is generated automatically in EndNote. We will do the formatting when the manuscript is accepted and no references have to be removed or added.*

Pg22 L4: "origin" should be "originate" I think

*OK*

Pg24 L10-11: are the "-" supposed to be there? Yes

Pg24 L18: "since we do know" – maybe just remove "do"

*ok*

Pg25 L5 and 12: extra parenthesis

*Reference format is generated automatically in EndNote. We will do the formatting when the manuscript is accepted and no references have to be removed or added.*

---

## Author Comment (AC4) · 30 Sep 2019

**Referee #3 reply**

**General comments**

This manuscript provides novel information on potential (de)nitrification and anammox rates combined with genomics in 6 abundant cold-water sponges, from which 5 have not been analyzed previously. The data show that denitrification is a common process in deepsea sponges and is relevant for understanding the role of sponges in nutrient cycling. The study seems well planned and conducted.

My main concern is that the potential denitrification rates measured in this study in tissue sections are upscaled to whole sponges and ecosystem level and are even used for future predictions under anthropogenic stress. The rates here should be treated as maximum or potential rates, since they were conducted with 10 times ambient concentrations, on small tissue sections in closed exetainers, under no oxygen and decreasing oxygen concentrations. *We agree that these are maximum potential rates. We make this now more clear in the discussion, beginning of chapter 4.4. Compare also comment from Referee #1, and our reply.*

My second major comment is that the MS is focused on and biased towards denitrification, with limited attention for other nitrogen transforming processes, such as nitrification, anammox and perhaps DNRA. I suggest to present all labelling incubations, carefully evaluate the results and present and discuss in a more balanced overview of the different nitrogen transforming processes and also include all data in the published Pangea dataset. *This work focusses on processes which transfer DIN into N2 – denitrification and anammox. The anammox rates were under detection limit. Coupled nitrification-denitrification is of interest in this context, but it was never our intention to provide "a balanced overview of the different nitrogen transforming processes" in sponges. Nitrification and also DNRA in sponges is well explored, while denitrification (and anammox) is not. Our intention was to close this knowledge gap.*

**Specific comments**
- Title: The term "nutrient sink" in the title is confusing and questionable
*We disagree and would like to keep it*
- P3, L3-11: this is too speculative, see major comment 1.
*We agree that this statement sounds provocative, but we still consider this a valid interpretation of our data. The interpretation is sufficiently justified in chapter 4.4.*
- P4, L16-19: This sentence doesn't really fit and perhaps the whole part of N fixation can be moved to the discussion, since it disrupts the introduction on DIN release.
*Section rephrased*
- P7, L20: Add some of the relevant characteristics for this site.
*The most relevant characteristic (hard bottom slope of a fjord) is given, details can be looked up in the quoted reference. Community structure of boreal vs arctic sponge grounds is now also described.*
- P2.2 and p2.3: A table or flow chart with the experimental incubations would be very useful.
*We consider the description in the text to be sufficient*
- P10, L:16: On the previous page it is mentioned that all incubations were done with water sampled from the deep, but here surface site water is mentioned for anoxic incubations. This needs to be clarified.
*Corrected for clarification*

- P11, L4: Can you be sure that 15-NO3 is reduced to 15-NO2, the preferred substrate for anammox? Please elaborate

*For the incubation experiment screening for anammox, 15-NH4 was added as substrate. 15-NO3 was added for the denitrification experiment. We realized that this was not clearly written in the method, this is now corrected.*

- P 11, L6-7: This is 1000% above ambient concentrations, and ambient concentrations from which site, arctic or boreal grounds or both?

*Section rephrased to avoid misunderstanding, see also Ref # 1*

- P11, L12: This is not *in situ* temperature for the Arctic species, the temperature increase might increase your potential rates.

*Lab experiments can never perfectly mimmick in situ conditions, we chose the best possible solution. We made a comment in the discussion that Arctic rates may be overestimated because incubation temperature was above in situ. See also comment by Ref #1.*

- Paragraph 2.3.2: There were no oxic sediment incubations?

*No. Since the denitrification was zero under anoxic conditions, there was not need to check for oxic conditions.*

- Paragraph 2.3.3: Including the calculations is informative for the reader.

*Some equations and calculations are now included.*

- P12,L22-P13, L1: The published dataset contains individuals with tissue degradation

*We now explained more clearly in chapter 2.3 that these samples were not considered for 15N analyses and rate quantification.*

- P13, L:11-13: This needs more explanation. I guess you mean no 29-N2 was detected in the anoxic incubations? What about 29-N2 and 30-N2 production in oxic incubations with labeled ammonium? Some production can be expected from coupled nitrification and denitrification.

*The lack of 29N2 production from 15NH4+ (anoxic incubations) suggests an absence of anammox, since N2 production via anammox requires 1 N from NO2- and 1 N from NH4+. It is also important to note that although labelled N2 production can be expected from coupled nitrification-denitrication of 15NH4+ in the oxic incubations, we did not detect labelled N2 in these oxic incubations.*

- Paragraph 2.3.4: Also here, the equations would be useful. And following my comment above, can you estimate coupled nitrification-denitrification from your oxic incubations with 15N-NH4?

*Equations added*

- Paragraph 2.4: In the results and discussion is mentioned that the sponges were also screened for anammox and N2 fixation functional genes. The screening and description of the functional genes should be described here. Was there also screening for other genes relevant for the nitrogen cycle (e.g. nitrification, DNRA)?

*Only screening for N2 fixation functional genes. This was relevant for our "nutrient sink" story, since nitrogen fixation would have closed the nitrogen cycle in the sponge. The other processes the reviewer mentions were of less importance for this particular story, and so we did not screen for genes.*

- P16, L1-6: Add graphs or tables with the results under oxic and anoxic conditions.

*These results are presented in Fig 2.*

- P16,L22-23: What about unlabeled N2 production?

*Unlabelled N2 production cannot be detected in the IR-MS.*

- P18, L5: I would remove "nutrient removal"

*This needs to stay, it is the main conclusion of the manuscript, and it is well justified.*

- P18, L22: Denitrification has not been directly shown in Fiore et al. 2013, but is given as a potential pathway, together with anammox or DNRA, to explain net consumption of nitrate in some of the sponges.

*Changed to "…has been indicated…"*
- P20, L6: I won't state that results are representative for normal conditions, but state that these conditions are not atypical (or something similar).
*We think this statement is well justified: the quoted literature proves that undersaturation of oxygen in the tissue is a common feature in sponges.*
- P20, L9-L15: I would expect year-round higher (dissolved) organic matter concentrations at the Boreal compared to Arctic grounds. Another explanation might be related to the higher incubation temperature (if it was 6°C) compared to the *in situ* temperature.
*Good point, text rephrased*
- Paragraph 4.2: The relevance of your denitrification rates in view of other nitrogen transforming processes should be discussed in a balanced way (see major comment 2). This paragraph gives the impression that denitrification is more important than nitrification in sponges, even though the majority of sponge studies reveal that sponges are net sources of nitrate, with denitrification being only a fraction of nitrification. Also the possibility of DNRA as competitive process for denitrification should be discussed somewhere.
*We removed the calculation of minimum nitrification rates as we see that this was confusing, and keep the statement that nitrification was present. We extensively quote publications which focus on the nitrogen source function of sponges. We do not believe that any reader will get the impression that denitrification is generally more important than nitrification in sponges. DNRA is not relevant for our publication as it conserves bioavailable nitrogen in the system. We focused on processes which remove bioavailable nitrogen from the system. As mentioned above, the aim of our study is not to give a balanced overview of the different nitrogen transforming processes in sponges, but to put the focus on sponges as potential nutrient sinks, and to point out potential scenarios based on calculated potential rates.*
- P21, L11-14: What is so different between explants and tissue sections? The tissue sections will also dependent on diffusion? There are more differences between Hoffmann et al. 2009, i.e. in your study you added $NH_4$, which will stimulate nitrification, while in Hoffmann et al. 2009, no $NH_4$ was added. You could discuss the reliability of nitrification measurements.
*Section removed and rephrased*
- P21, L21-25: "May be higher" should be "are likely higher" and the reported rates are really at the low end of other reported rates.
*Section rephrased*
- P22, L1, yes, but you added 10 μM (unlabeled) $NH_4$, which can result in 10 μM (unlabeled) $NO_3$ in oxic conditions.
*Labelled ammonium was added to the annomox incubations. No ammonium was added to the denitrification experiment. We now discovered that we did not write this clearly in the method section, which has led to confusion. The text is now corrected*
P22,L6,9: These last two sentences are not connected to the rest of the paragraph.
*Yes they are. This paragraphs lines up several facts that lead to the main conclusion that the nitrogen cycle is not closed in these sponges, so bioavailable nitrogen leaves the system.*
- P22, L16-21: I won't use optimized, I guess you want to say there is an active denitrifying community. Perhaps add some statistics to the relationship, this is a nice result.
*Rephrased.*
- P23, L4-6: I disagree that they are realistic, see major comment one.
*Rephrased, but see our reply to major comment one.*
- P23, 15-17: Are there reported denitrification measurements of Arctic sediments? What about the other nitrogen transforming processes? A comparison to literature should be added with to this statement.

*There is a recent publication about anammox in these Arctic sediments:*
*https://www.biorxiv.org/content/10.1101/729350v1 , where profiles of nitrate, ammonium and oxygen are included. The group did also predict denitrification rates based on their model but these data are not yet published. Apart from this, we are not aware of any literature on denitrification rates in deep Arctic sediments, only from the continental shelf areas.*

- Paragraph 4.4: The results are not representative for a natural situation, but rather show a potential, so I would be extremely cautious to upscale these numbers and refer to these sponges as efficient nutrient sinks (see major comment 1).

*We rephrased the first paragraph to make more clear that we talk about maximum potential rates and possible scenarios. We still think that a presentation of (potential) areal rates is valid and useful to compare denitrification rates of sponge grounds to other ecosystems.*

- P24, L6-10: The calculations and conversion factors going from volume to surface integrated measurements are lacking (but see major comment 1).

*No, both calculations and conversion factors are given in the data publication https://doi.pangaea.de/10.1594/PANGAEA.899821 which will be publicly available when the manuscript is accepted.*

- P25, L13-15: Combined anthropogenic stressors can also lead to changes in nutrient and organic matter availability which might affect microbial composition and biogeochemical processes. It is too speculative to state that sponges will become nutrient sinks in the future if they reduce pumping.

*We say "may", not "will". We think it is necessary to point out this potential and so-far overlooked scenario.*

- P26, L10: Please expand the dataset in Pangaea with all incubation data and results.

*All results of the incubation experiments are completely presented in terms of 29N and 30N accumulations. What is lacking?*

- P35, L2: STP is used for the first time, does it stand for standard temperature and pressure?

*Removed, as this explanation belongs to the methods and not the figure legend*

- Figure 1 and figure 2 should be swapped, based on their reference order in the text.

*No, Figure 1 is mentioned for the first time on page 12, while Figure 2 is mentioned for the first time on page 16.*

- The order of references in the citations and full references need to be checked.

*Reference format is generated automatically in EndNote. We will do the formatting when the manuscript is accepted and no references have to be removed or added.*

---

## Author Comment (AC5) · 1 Oct 2019

See supplement

Please also note the supplement to this comment:
https://www.biogeosciences-discuss.net/bg-2019-135/bg-2019-135-AC5-supplement.zip

———————————————————

---

## Referee Report (RR1)

**Review revised version Rooks et al. 2019 – reviewer 3**

The authors adjusted and improved the manuscript according to most the reviewers comments. The supplementary pdf with track-changes was also useful to check the adjustments. However, not all previous reviewer comments have been sufficiently addressed. In my (professional) view, the authors should be more careful in upscaling and translating the potential denitrification rates to ecosystem level.

**Title:** Sink is a confusing term, at least to me; and also "high" is vague, "high" compared to what? Why not change it to (something like): "Nutrient removal through denitrification is a common feature of boreo-arctic sponges."

**Paragraph 4.4:** If you want to upscale to *in-situ* conditions, the rates should be adjusted to ambient nitrate concentrations. The easiest way is a linear relation (first order kinetics) between nitrate concentrations and denitrification rates, resulting in 10 times lower denitrification rates under ambient nitrate concentrations. If you assume that nitrification is not driven by nitrate concentrations, then this should be discussed and justified.

**P26, L14-16:** pumping and non-pumping rates are swapped for the Arctic sponges.

**P26: L19:** add under anoxic conditions. As also discussed in the paragraph below and I agree with this discussion, this value is based on theoretical extremes. So is there any ecosystem relevance to this number? I would also remove this from the abstract.

**Table 1:** Could you add a column with denitrification rates under anoxic conditions as well? Next to visualization of the data in Fig.2, t is useful to have all numbers presented in text and tables, so these numbers can be used by others. I commented this before. I would also use a similar order of sponges in table and figure 2 (*Parva* and *Hentscheli* are swapped)..

---

## Author Response (AR2)

Dear editor, dear Jack,

Please find our replies to all reviewers' comments attached, our replies in italic font. We included most of the reviewers' suggestions to further improve the manuscript. We hope that the revised version now may be acceptable for publication in Biogeosciences.

There are still some mistakes in the format of the in-text citations and the reference list due to the auto-format function in the EndNote program, which we were not able to switch off. The citations and reference list are complete and we can easily fix this issue as soon as we have understood how.

Again we would like to thank you for your fair and competent review process.

With best regards,
On behalf of all co-authors,
Friederike Hoffmann

**Report #2**

Submitted on 29 Nov 2019
Anonymous Referee #1

**Anonymous during peer-review: Yes** No

**Anonymous in acknowledgements of published article: Yes** No

**Recommendation to the editor**

| | |
|---|---|
| **1) Scientific significance**
Does the manuscript represent a substantial contribution to scientific progress within the scope of this journal (substantial new concepts, ideas, methods, or data)? | Excellent **Good** Fair Poor |
| **2) Scientific quality**
Are the scientific approach and applied methods valid? Are the results discussed in an appropriate and balanced way (consideration of related work, including appropriate references)? | Excellent **Good** Fair Poor |
| **3) Presentation quality**
Are the scientific results and conclusions presented in a clear, concise, and well structured way (number and quality of figures/tables, appropriate use of English language)? | **Excellent** Good Fair Poor |

For final publication, the manuscript should be

**accepted as is**

accepted subject to **technical corrections**

**accepted subject to minor revisions**

reconsidered after **major revisions**

> I am willing to review the revised paper.

> I am **not** willing to review the revised paper.

**rejected**

**Suggestions for revision or reasons for rejection (will be published if the paper is accepted for final publication)**

The manuscript has been revised substantially and my major comments to the original version have been taken into account. The revision is a substantial improvement, and the manuscript now conveys clear and justified conclusions. Importantly, the central results from 15N incubations have been revised in detail, with the discovery of critical errors in the original calculations. The revised results now appear completely plausible. I have suggestion for slight rewording, and a few points need further clarification.

p. 3 l. 4-8: "Scenario" here and later in the text is used in way that doesn't fit with my understanding of the word, and since the authors now emphasize the potential nature of the measured rates, "scenario" is not needed. Suggestion: "The results suggest a high potential denitrification capacity… and Arctic sponge grounds, with areal denitrification rates up 0.6…for non-pumping sponges and even 0.3 mmol N m-2 d-1 for pumping sponges."

*We agree that "scenario" has a distinct definition at least in the climate change community which indeed does not fit here. We rephrased throughout the text according the reviewer's suggestion.*

p. 5 l. 4-6: Nitrification and denitrification don't compete – what would they compete for?
*Section rephrased*

p. 13 l. 20- p.14 l. 1 + 10-12: I am confused here: First you say that samples showing an abrupt increase were excluded even though they didn't show visual signs of degradation, and then you say that increases represent incipient anoxia BECAUSE there were no signs of degradation – if samples with such increases were excluded, why do we hear about them later, and how can visual inspection be trusted in the 2nd case when it couldn't in the 1st.

*We now rephrased: "Though we never observed **visual** signs for tissue degradation (see for example (Hoffmann et al., 2003;Osinga et al., 2001;Osinga et al., 1999) for description of how to spot signs of sponge tissue degradation), some samples showed an abrupt increase in $N_2$ production, **indicating the onset of degradation."***
*We do not hear about these samples later, we clearly state **"These were not included in the analyses and rate calculations".***

p. 16 l. 21: This seems to repeat l. 13 (and 15N isotopes were not produced). Also, please mention that the rates reported are D14 (at least I think they are?).

*We were not able to identify the text part which the reviewer refers to. But see changes we made in chapter 2.3. where this comment most likely refers to.*

p. 19, l. 16: Instead of "0" I suggest "below detection" (and "n.d." for the tables).

*Ok, changed accordingly*

p. 20 l. 7: italics missing

*ok changed*

p. 24 l. 19: 0 instead of O

*We were not able to identify the text part which the reviewer refers to.*

p. 26 l. 24: In their rebuttal the authors boldly state that this is textbook knowledge and refuse to add a reference. I would be hesitant with such statements (or acknowledge that textbooks can be wrong). See, e.g., Jones et al. (2008) Mol. Biol. Evol. 25:1955-66.

*We removed the sentence as it is not critical for our story*

p. 28 l. 10: Instead of "scenarios about the" how about "first-order estimates of"?

*Section rephrased*

Also, considering that numbers here are rough estimates, I would use just 1 or 2 significant digits.

*We think 3 digits are ok.*

Table 1: The number of significant digits should reflect the precision of the data.

*We agree*

**Review revised version Rooks et al. 2019 – reviewer 3**

The authors adjusted and improved the manuscript according to most the reviewers comments. The supplementary pdf with track-changes was also useful to check the adjustments. However, not all previous reviewer comments have been sufficiently addressed. In my (professional) view, the authors should be more careful in upscaling and translating the potential denitrification rates to ecosystem level.

**Title:** Sink is a confusing term, at least to me; and also "high" is vague, "high" compared to what? Why not change it to (something like): "Nutrient removal through denitrification is a common feature of boreo-arctic sponges."

*We would like to keep "sink" as we consider this term appropriate in this context. We see the issue concerning "high", and suggest changing the title to: "Deep-sea sponge grounds as nutrient sinks: Denitrification is common in boreo-arctic sponges".*

**Paragraph 4.4:** If you want to upscale to *in-situ* conditions, the rates should be adjusted to ambient
nitrate concentrations. The easiest way is a linear relation (first order kinetics) between nitrate concentrations and denitrification rates, resulting in 10 times lower denitrification rates under ambient nitrate concentrations. If you assume that nitrification is not driven by nitrate concentrations, then this should be discussed and justified.

*The presented denitrification rates are estimated to represent ambient NO3 concentrations as described in eq 1-4. This has now been clarified in the following sections:p16, l8-10; p19 l7; p20 l5.*

**P26, L14-16:** pumping and non-pumping rates are swapped for the Arctic sponges.

*OK, corrected*

**P26: L19:** add under anoxic conditions.

*"Anoxic conditions" will be wrong in this context, it is not the oxygen condition in the ambient water but those in the sponge tissue (as a result of stopped pumping activity) which is the point here. We rephrased accordingly.*

As also discussed in the paragraph below and I agree with
this discussion, this value is based on theoretical extremes. So is there any ecosystem relevance to this number? I would also remove this from the abstract.

*We removed these numbers from the abstract as these are indeed not our core results. But we would like to keep them in the discussion to make people think about future potential ecosystem consequences.*

**Table 1:** Could you add a column with denitrification rates under anoxic conditions as well? Next to visualization of the data in Fig.2, t is useful to have all numbers presented in text and tables, so these numbers can be used by others. I commented this before.

*We commented before that all original data can be downloaded from Pangaea in the form of excel-tables to be re-used by others. Nevertheless, we now also included denitrification rates under anoxic conditions in Table 1.*

I would also use a similar order of sponges in table and figure 2 (*Parva* and *Hentscheli* are swapped).

*OK, also S. fortis and G. atlantica were swapped, both changed.*

[revised manuscript text omitted]

---

## Author Response (AR3)

**Dear editor, dear Jack,**

We are pleased to hear that our manuscript is now accepted for publication. We have applied all technical revision as requested in your letter below. The in-text reference formatting has been corrected as requested.

Best regards,
Friederike Hoffmann

All technical corrections were applied as requested below:

All through, the reference formatting needs attention; the references should be chronologically when part of a series).

p. 4, line 5: year of Brusca and Brusca reference is missing

All through, use consistently Arctic rather than arctic.

p. 8, line 14: reformulate: from less than 1 and 2 mg L-1… does not make sense.

p. 10, line 9. Floating , and .

p. 13, line 8-10. This is unclear. Did you run 3 samples per sponge species for sediments?

p. 14, line 9: delete 'in these'

p. 19, line 8: delete in ch. 2.3.

p. 21, line 4: six key sponge species

p. 22, line 17: …section), because we did not have control…

p. 23, line 9: replace O with 0 (already alluded to by referee).

p. 23, line 13-16: do not use bold to emphasize conclusions
same applies to page. 24, line 12-13

p. 24, line 4: and should not be italic

p. 24, line 25: 15N should be superscript

p. 26, line 5, lin3 9, line 16: do include the units!

p. 27, line 17: reference list issue

p. 28, line 5: delete main

p. 29, line 11: Measurements… were done…

Table 1: As the referee suggested, lower number of digits, they are not significant.
Table 1. Explain ND (as Not Detected).

Table 1: units per sponge is not right for sediment